

# Seasonal cycles of the carbon export flux in the ocean: Insights from the SISSOMA mechanistic model

Athanasios Kandylas[1] and Andre William Visser[1]

[1]VKR Centre for Ocean Life, National Institute of Aquatic Resources, Technical University of Denmark, Kongens Lyngby, Denmark

**Correspondence:** Athanasios Kandylas (athka@aqua.dtu.dk)

**Abstract.** This study aims to investigate the seasonal dynamics of carbon export flux in the ocean using the SISSOMA modeling framework. SISSOMA uses a 2-dimensional state space (size and excess density) to follow the fate of aggregates in the mixed layer which are transformed through three main processes, e.g., aggregation, fragmentation, and remineralization, until they eventually sink out of the surface ocean. The model tracks aggregate size, mass, and porosity which allows for a direct estimate of aggregate sinking speed through a Reynolds number modified Stokes' law. First, a simple seasonal cycle with a single peak of POM production is presented, which provides a solid basis to understand the model's dynamics and enables us to perform sensitivity analyses on important parameters. The effect of increased stratification on a reconstructed ecosystem in the north Atlantic is then presented and discussed. Overall, our results showcase the nonlinear relationship between the production of primary particles and the export of aggregates out of the mixed layer and unveil key mechanics of the three transformative processes. Moreover, it has been shown that remineralization rates, stickiness, and the size/ excess density characteristics of the primary particles all affect in various ways the intensity, seasonal cycle, and the resulted size spectrum of the aggregate community. Finally, our results indicate the crucial role that turbulence plays in both the timing and the magnitude of the carbon export flux which might affect not only the potential of the system to remove carbon out of the mixed layer but also have a direct impact on the organisms inhabiting the mesopelagic layer which rely on the sinking particles to cover their energetic needs.



## 1 Introduction

The production and export of particulate organic matter (POM) from the surface ocean to depth constitutes an important pathway in the biogeochemical cycle of the ocean, maintaining in part the carbon stored by the biological carbon pump (BCP) (Boyd et al., 2019; DeVries, 2022), as well as supplying benthic and mesopelagic ecosystems with organic carbon and energy (Billett et al., 1983; Hernández-León et al., 2020). It is qualitatively well understood that export is ultimately governed by primary productivity in the sunlit surface ocean. Although a mechanistic description of how these are related remains elusive (Boyd and Newton, 1995; Buesseler, 1998; Henson et al., 2012), a simple heuristic measure, the $e-ratio$ (the ratio of export production $F$ to net primary production $P_{NPP}$, sometimes referred to as export efficiency) is often invoked in models and analyses (Siegel et al., 2014; Henson et al., 2019; Buesseler et al., 2020; Laurenceau-Cornec et al., 2023). Factors that contribute to the variation in $e-ratio$ are myriad and include the structure of the plankton community (Boyd and Newton, 1995; Guidi et al., 2016; Henson et al., 2019), nutrient cycling (Kemp et al., 2000; Raven and Waite, 2004), the turbulence and stratification, and cell exudates (Passow et al., 2001; Engel, 2004). A somewhat simplifying perspective can be achieved by dividing the $e-\mathrm{ratio}$ into two components; the first being the relationship between $P_{\mathrm{NPP}}$ and $P_{\mathrm{POM}}$, the production rate of primary particulate organic material (Laufkötter et al., 2016). This component ($P_{\mathrm{POM}}/P_{\mathrm{NPP}}$) is largely controlled by the dynamics of the plankton population and is generally the most easily accessible component of the export flux simulated in ecosystem models. The second component is the relationship between POM production and export, $F$. This second component is largely controlled by aggregation processes, transforming primary detrital material into fast sinking aggregates, and it is here that perhaps the greatest uncertainty lies in estimating export flux. The overall estimation of the e-ratio can thus be broken down as:

$$e-\mathrm{ratio} = \frac{F}{P_{\mathrm{NPP}}} = \frac{P_{\mathrm{POM}}}{P_{\mathrm{NPP}}}\frac{F}{P_{\mathrm{POM}}} \tag{1}$$

(Laufkötter et al., 2016). It is the second factor in this relationship (termed the $s-\mathrm{ratio}$ in Laufkötter et al. (2016)) that our model SISSOMA (Visser et al.) seeks to resolve and is the focus of this study.

Given the production rate and characteristics (e.g. size, ballasting, mass) of primary POM particles, we simulate the formation and transformation of POM into porous aggregates while estimating sinking speed directly via Stokes' law corrected for a finite Reynolds number. We simulate not only aggregation, but also degradation and fragmentation, two important but often poorly resolved aspects of aggregate dynamics. In particular, the sinking velocity of the resulting aggregates is crucial for their fate, as fast-sinking aggregates are subjected to lower degradation loses by bacterial remineralization in the surface and eventually reach greater depths in the oceans' interior. The aggregate's sinking velocity depends on its size, excess density, composition e.g. ratio of organic matter to ballast minerals) and porosity, while degradation rates are mainly defined by the aggregate's lability, the structure of the microbial community and temperature (Baumas and Bizic, 2024a). Finally, it is well understood that the relationship between $P_{\mathrm{POM}}$ and $F$ exhibits a strong nonlinearity (Kiørboe et al., 1994). This is qualitatively described as a critical concentration (Jackson, 1990; Burd and Jackson, 2009) where aggregation to larger fast sinking aggregates accelerates rapidly and can lead to episodic intense export events (Siegel et al., 2024) precipitated by intensifying turbulence. It is far from clear that any simple relationship exists between $P_{\mathrm{POM}}$ and $F$, and by extension $F$ and $P_{\mathrm{NPP}}$. This





is reflected in the high variability in the observed $e-$ratio (Henson et al., 2012, 2019; Siegel et al., 2023) both spatially and temporally, suggesting that a more mechanistic analysis is required.

A particular complicating feature is that these dynamics are very rarely in steady state. Over much of the world's oceans, seasonal cycles of productivity imprint their effects throughout the marine ecosystem (Visser et al., 2020), including POM dynamics (Stramska, 2009; Laufkötter et al., 2016) and export (Laurenceau-Cornec et al., 2023). Subtropical ecosystems are

relatively stably stratified throughout the year, nutrient limited and dominated by the microbial loop, whereas temperate latitude seas such as the north Atlantic are characterized by large fluctuations in many physical processes and abiotic factors, such as mixing, light availability, and temperature. These factors, in turn, play a crucial role in the resulting plankton community with a relatively stable seasonal cycle and the dominance of small cells in the case of oligotrophic (low latitude) ecosystems and distinct, high-growth phases, e.g., spring and autumn 'blooms', of diatoms and larger calanoid copepods at higher latitudes

(Visser et al., 2020). Overall, it has been suggested that the total production, size-structure of POM and the eventual export of material out of the mixed layer is directly connected to the plankton community structure, as well as the physical forcing. Additionally, ongoing climate change not only changes export patterns (Brun et al., 2019) but also increases the complexity of the system by shifting the balance in hard to predict feedback loops. Increasing temperature, for instance, might have both a positive and negative effect on the export flux, by decreasing the water density and hence the particles' sinking speed, on

the one hand, and by increasing the metabolic rates of microbes and the consequent remineralization rates of the sinking particles, on the other hand (Henson et al., 2022). In understanding and modeling these processes, it is important to maintain as mechanistic an approach as possible in order to provide more reliable predictions of global biogeochemical cycles under accelerating climate change conditions.

In this paper, we explore the seasonal dynamics of POM in the upper mixed layer of the ocean by following its journey

from its production to its export. We approach this mechanistically using the SISSOMA modeling framework (Visser et al.) and reconstructing characteristic seasonal POM production cycles. Through a variety of sensitivity analyses, we try to reveal the key mechanisms of the export system and investigate how it is affected by certain factors, e.g., remineralization, stickiness, size, and excess density characteristics of the primary particles, as well as understand the effect of turbulence on the export system dynamics.

## 75  2  Methodology

### 2.1  Model description

SISSOMA is a mechanistic model that tracks the fate of aggregates through three main processes, i.e., aggregation, fragmentation, and degradation, and their consequent sinking (Visser et al.). These processes are simulated in a 2-dimensional state space, size and excess density of the aggregates. The model accounts for the variation of aggregate porosity, mass, and size, allowing

for a mechanistic estimate of the aggregate sinking speed while they are constantly being redistributed in the 2-dimensional state-space through transformative processes. The aggregation model can use as input a description of the formation of primary particles, representing for instance dead and dying plankton cells, fecal pellets, and/or aolean dust deposits which are all char-





acterized by their size and excess density. Many of the key characteristics of primary particles can be linked to their origin, for instance diatoms will have a higher excess density than other phytoplankton taxa (Hansen and Visser, 2019), and the size of

the fecal pellets will be governed by the size of zooplankton grazers (Stamieszkin et al., 2015). For the purposes of this study, we construct primary particle production rates that can be inferred from typical seasonal cycles of plankton dynamics observed in nature. In principle, however, the model can be coupled directly to any suitable size and trait-resolved plankton model.

The physical space representation of SISSOMA is kept as simple as possible: a uniform surface mixed layer, although this can be expanded to multiple layers stacked to each other. The state space representation, where the particle dynamics takes

place largely, represents the size (radius) $r$ and excess density $\rho'$ of the emerging aggregate community. SISSOMA provides flexibility to choose the range and the number of classes of this space. For this project, the size of possible aggregates in the system, $r$, ranges between 1 and 1E6 $\mu$m and consists of 30 logarithmically spaced bins ($x$), while their excess density, $\rho'$, extends between 1.6E-6 and 64.2 kg m$^{-3}$ ($y$), Fig. 1(a). This information is then used to calculate the sinking speed matrix, Fig. 1(b), by utilizing a modified version of Stokes' law:

$$w^2 = \frac{8}{3} \frac{\rho'}{\rho_w} \frac{g\,r}{C} \tag{2}$$

where $\rho_w$ is the density of the surrounding water, $C$ is the drag coefficient, and $g$ is the gravitational acceleration.

A rigorous mathematical description of particle aggregation dates back over a century (Smoluchowski (1918)). Following more recent formulations (Burd and Jackson, 2009; Jokulsdottir and Archer, 2016) and including specific representations for

fragmentation and degradation, aggregate dynamics can be summarized as:

$$\frac{\partial}{\partial t}\mathsf{N}(\mathsf{s},t) =$$

$$q_{m:c}P_{\mathrm{POM}}(\mathsf{s},t)$$

$$+ \frac{\alpha}{2} \int_{\mathsf{S}} \beta(\mathsf{s}',c^\star(\mathsf{s},\mathsf{s}'))\,\mathsf{N}(c^\star(\mathsf{s},\mathsf{s}'),t)\,\mathsf{N}(\mathsf{s}',t)\,d\mathsf{s}'$$

$$- \alpha\,\mathsf{N}(\mathsf{s},t) \int_{\mathsf{S}} \beta(\mathsf{s},\mathsf{s}')\,\mathsf{N}(\mathsf{s}',t)\,d\mathsf{s}'$$

$$- \psi(\mathsf{s})\,\mathsf{N}(\mathsf{s},t) + \int_{\mathsf{S}} \psi(\mathsf{s}')\,\mathsf{N}(s',t)\,\theta(s,s')\,ds'$$

$$- \frac{\partial \mathsf{s}}{\partial t} \cdot \frac{\partial \mathsf{N}}{\partial \mathsf{s}}$$

$$- \mathsf{N}(\mathsf{s},t)\,\frac{w(\mathsf{s})}{h} \tag{3}$$

where $\mathsf{N}(\mathsf{s})$ is the number of aggregates in a given state-space bin, $\mathsf{s} = (r,\rho')$ is a new state variable which includes all aggregate attributes and the function $c^\star$ defines the state of a parent aggregate based on the state of both the other parent and the produced-daughter aggregate. Primary particles enter the mixed layer at a rate of $q_{m:c}P'_{\mathrm{POM}}$, where the first term represents the dry mass

to carbon ratio and the second is the time and state-resolved production rate of POM. That is, $P'_{\mathrm{POM}}d\mathsf{s}$ is the production rate





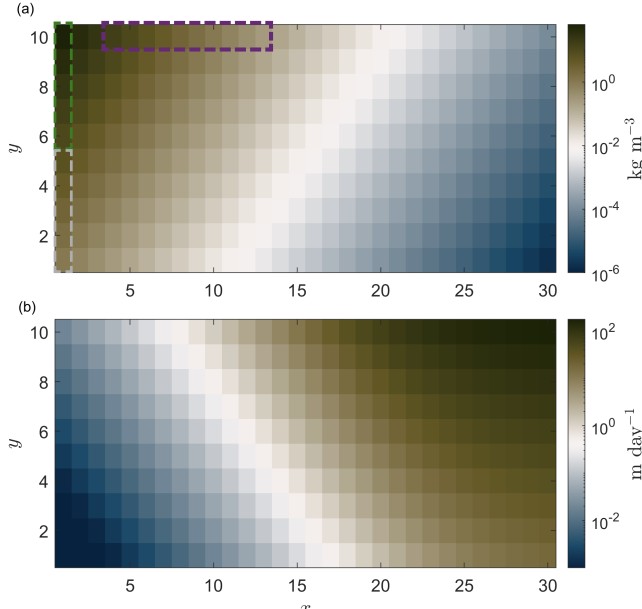

**Figure 1.** Representation of the (a) aggregate excess density, $\rho'$, and (b) sinking speed, $w$, in the 2-dimensional state space where $x$ and $y$ are the scaling factors. a) Larger aggregates lead to lower excess densities, as surrounding water is trapped in their interior during the formation period. The colors point out the three scenarios in the sensitivity analysis in section 3.2, where different size/ excess density primary particles are introduced in the system: i) small and light (S+L) in grey, small and dense (S+D) in green, and big and dense (B+D) in purple. b) the sinking speed of aggregates as derived from Eq. (2).

of POM dry mass in the same interval of state space $[\mathsf{s}, \mathsf{s} + d\mathsf{s}]$. Note that $P_{\mathrm{POM}} = \int_{\mathsf{S}} P'_{\mathrm{POM}} d\mathsf{s}$, i.e. the integral over all state space.

The aggregation process is captured by the following two terms and relies both on the encounter kernel of two particles, $\beta(\mathsf{s}, \mathsf{s}')$, and their stickiness, $\alpha$, which defines the probability of two particles to form an aggregate once they have collided. The larger an aggregate, the higher the chances that it breaks down at a rate of $\psi(\mathsf{s})$, and is redistributed into smaller with variable excess density aggregates according to the partitioning function $\theta(\mathsf{s}, \mathsf{s}')$. The next term describes loses from remineralization, which we assume that it occurs at a constant rate throughout the state space and removes only dry mass leading to aggregates of the same size-class but reduced excess density. Finally, whether aggregates are exported out of the mixed layer, $h$, depends on their sinking speed, $w(\mathsf{s})$.

## 2.2 Analyses description

Two distinct analyses that focus on the seasonal cycle of POM production, the properties of the primary particles, and the prevailing environmental conditions are performed. In the first part of the analysis, a highly seasonal environment is reproduced with a cycle of primary particle production peaking in the middle of April at 1000 mgC m$^{-2}$ day$^{-1}$ , Fig. 2(a). For simplicity,



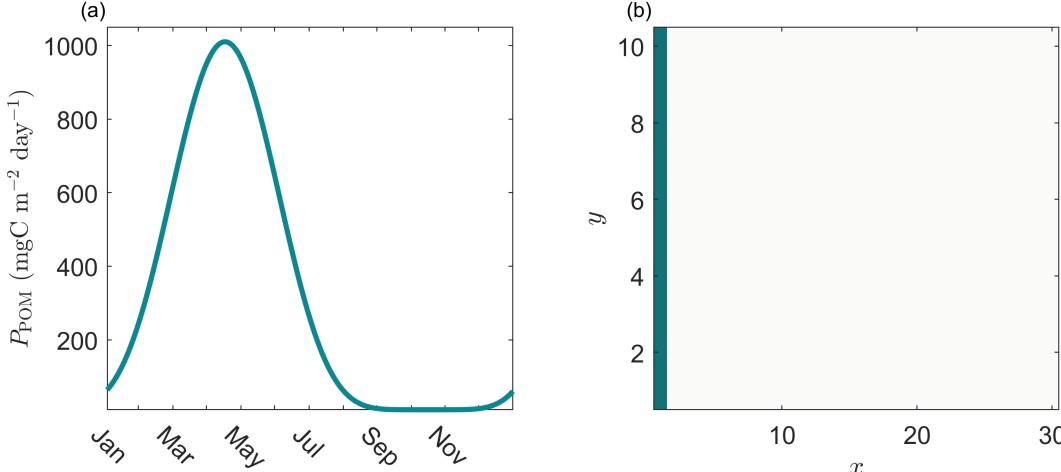

**Figure 2.** Reconstruction of (a) a simple annual cycle of primary particles peaking in middle April, (b) which are assigned to the smallest size class ($x$) and evenly distributed throughout the excess density space ($y$) which form the base for the results in sections 3.1 and 3.2.

we allocate the primary particles to the smallest size class and distribute them equally throughout the excess density field, Fig. 2(b). This allows us to better understand both the dynamics of the system, section 3.1, given a set of parameters (see in the Appendix, Table A1) and the effect of remineralization, stickiness and the primary particles' size/ excess density characteristics on the export flux, section 3.2. Regarding the latter, we simulate three scenarios based on the relative characteristics of the primary particles: a) small and light, S+L (in grey), b) small and dense, S+D (in green), and c) big and dense, B+D (in purple) primary particles, as shown in Fig. 1(a). Further, we investigate the effect of episodic storm events on export flux with regards to their intensity (i.e. the level of turbulent dissipation rate), their duration, and their occurrence in different phases of the seasonal cycle, see Section 3.2.

In the second part of the analysis, a unicellular plankton community in the north Atlantic is replicated where a strong 'spring bloom' and a weaker 'autumn bloom' are observed. In this case, we distinguish between primary particles that originate from diatoms and the rest of the unicellular organisms (i.e. bacteria, phytoplankton, heterotrophic microzooplankton and mixotrophs). For simplicity, we term the latter as generalists (Cadier et al., 2020; Serra-Pompei et al., 2020), Fig. 3(a). In the beginning of the 'spring bloom, the optimal light, temperature and stratification conditions, as well as the high availability of nutrients, give a competitive advantage to diatoms. They grow faster until a point where nutrients (especially silicate) become limited and the rest of the unicellular organisms take over as their smaller sizes enables them to diffuse sparse nutrients into their cells more efficiently. The same pattern but in lower magnitude is reproduced in the case of the 'autumn bloom'. Regarding the distribution of the primary particles in the 2-dimensional space, we assume that when an organism dies produces POM of the same size. Diatoms' radius ranges between 3.2 and 50.6 $\mu$m, whereas the rest of the unicellular organisms' radius varies between 1 and 32 $\mu$m. Moreover, in the base that diatoms posses a silica shell and their excess density is higher than the rest of the unicellular community, we distribute the primary particles originating from diatoms in the upper half of the excess density





space while primary particles introduced by the rest of the unicellular organisms are equally allocated throughout the full length

of the excess density space, Fig. 3(b). Finally, we investigate how the mixing conditions in the upper 100m of this environment affect the transformative processes of POM (namely aggregation and fragmentation) and the resulted carbon export. This is accomplished by focusing on the turbulent dissipation rate through two scenarios: one with high seasonal variability and a second scenario with a relatively constant rate, Fig. 3(c).

Throughout the report, three different annotations are used to describe the export flux (mgC m$^{-2}$ day$^{-1}$): a) $f_{100}$ refers to

the density integrated export flux (Eq. 4), b) $F_{100}$ is the total export flux in each point in time (Eq. 6) and c) $F_{tot}$ is the total export flux in the span of one year.

$$f_{100} = \frac{1}{h}\sum_y m_{\mathsf{dry}}(x,y)w(x,y) \tag{4}$$

$$F_{100} = \frac{1}{h}\sum_{x,y} m_{\mathsf{dry}}(x,y)w(x,y) \tag{5}$$

where $h$ is the mixed layer depth and $m_{\mathsf{dry}}$ is the total dry mass concentration (mgC m$^{-3}$) and $w$ is the sinking velocity matrix (m day$^{-1}$). Finally, following (Laufkötter et al., 2016) we define the $s-\mathrm{ratio}$ as the fraction of the total export of material in each time ($F_{100}$) to the introduction of new, primary particles ($P_{\mathrm{POM}}$) in that time:

$$s-\mathrm{ratio} = \frac{F_{100}}{P_{\mathrm{POM}}} \tag{6}$$

One aspect of productivity-aggregation-export dynamics that is particularly revealing is the phase relationship between the

different processes (Wassmann, 1997; Laufkötter et al., 2016; Laurenceau-Cornec et al., 2023). In general, export production for any given system will describe an open orbit when plotted against either net primary production ($P_{\mathrm{NPP}}$) or POM production ($P_{\mathrm{POM}}$). It has been proposed that systematic patterns in the shape of these orbits can be leveraged to provide more nuanced information on the $F_{100}$-$P_{\mathrm{NPP}}$ relationship on regional scales (Laurenceau-Cornec et al., 2023). Establishing a solid understanding along these lines would come a long way in linking synoptic satellite observations with global biogeochemisrtry. With

these concepts in mind, we will analyses $F$-$P_{\mathrm{POM}}$ phase relationships to better understand the mechanisms behind some of these systematic patterns.

## 3 Results

### 3.1 Model mechanics

In the first part of the results, the analysis on a simple seasonal cycle where small primary particles with variable excess densities

enter the mixed layer is presented. Fig. 4(a) shows how both the production of new material and the density-integrated export flux ($F_{100}$) evolve in time, as well as it illustrates the instantaneous ratio between the export flux to the production of new particles ($s-\mathrm{ratio}$). The input of new material in the system begins on day 297 (end of October) and peaks on day 107 (middle





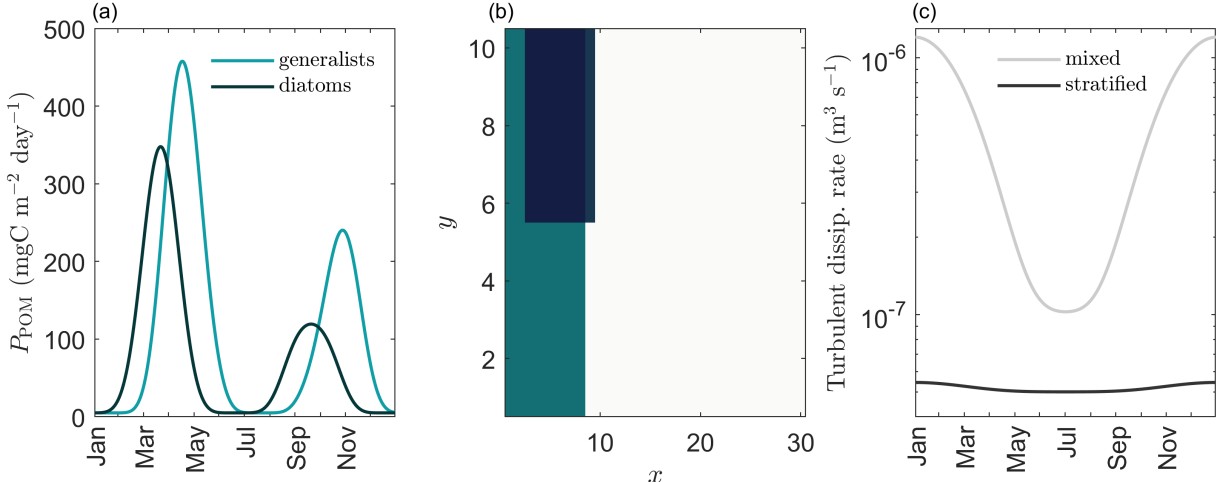

**Figure 3.** Reconstruction of an ecosystem in the north Atlantic with primary materials originating from two functional groups, i.e., diatoms and generalists, used for the analysis in section 3.3. a) The annual cycle is characterized by two distinct 'bloom' phases where new material enter the system, b) primary particles originating from diatoms are assigned to larger size bins and are equally distributed to higher excess density ranges compared to generalists, c) two scenarios were used to represent the stratification conditions of the system: a highly seasonally-mixed and a stratified throughout the year.

of April), whereas the export of aggregates out of the mixed layer begins on day 62 (early March). This implies that there is a time-lag of 130 days between the first/ primary particles entering the system and the first aggregates being exported out of the

mixed layer. Focusing on the evolution of the exported flux in time, from day 62 (early March) to day 124 (early May), 62 days, there is a 'preparation phase' of a very weak response. By referring to Fig. 4(b), during this period there is an accumulation of small particles in the system which then leads to the production of progressively bigger aggregates. This accumulation of mass does not lead yet to a strong export flux which can be explained by the fact that these first aggregates are small, in the range of $1\mu$m to $200\mu$m, and light meaning that they sink very slowly.

However, over the next 9 days (day 125 to day 134) there is a distinct shift in the observed size-spectrum of the dry mass in the system creating aggregates with almost five orders of magnitude of difference in their size range, Fig. 4(b). This implies that there is a critical concentration of mass where the aggregation process really kicks off producing aggregates of variable (and bigger) sizes and excess densities which in turn are able to support this 'exponential phase' in the total amount of exported material, Fig. 4(c). The following period until the peak of the export flux on day 186 (early July) at 281.8 mgC m$^{-2}$ day$^{-1}$

shows a more gradual increase in the total amount of exported material and is dominated by aggregates of sizes around $10^4$ $\mu$m. An interesting point of the dynamics of this system is how the export flux is sustained even after the entry of new material is diminished, especially the period between day 181 (start of July) and day 347 (early December) where the export flux is higher than the production, Fig. 4(a). This can be explained by the fact that on the one hand the system has still enough material to keep supporting the formation of optimal-sinking velocities aggregates and, on the other hand once big aggregates, which are

**Figure 4.** System dynamics for a simple annual cycle where the smallest size-class primary particles with variable excess densities enter the mixed layer. a) The progression of the production, total export flux and instantaneous $s-$ratio in time, b) the distribution of dry mass in the size-spectrum in each time-step, c) the annual evolution of the density integrated export flux, ($f_{100}$), and d) the system-characteristic phase diagram where the total export flux ($F_{100}$) is plotted against the POM production ($P_{POM}$).





not heavy enough to sink, reach the fragmentation's size threshold, they break down into smaller aggregates of variable excess densities. This in turn fills the 2 dimensional space via two directions (small primary particles and fragments of the biggest and light aggregates), increases the diversity of the aggregate properties in the system and increases the chances for new aggregates with the optimal sinking speeds to be formed. In contrast to this, reminralization has also an increasingly leading role during this period by moving mass to lower density bins which eventually leads to the diminishing of the export flux.

Finally, the phase diagram in Fig. 4(d) highlights the strong non-linear relationship between the production of new material and the resulted export flux. The black dot indicates the annual mean fluxes of production and export which results in an annual mean $s-$ratio of 0.32. In extension to this value, Fig. 4(a) shows how $s-$ratio fluctuates widely throughout the year with a minimum of 0.0015 on day 71 (middle March) and a maximum of 15.62 on day 250 (early September).

## 3.2 Sensitivity analysis

In the second part of the analysis of this system, a set of sensitivity analyses on three parameters, e.g., remineralization, stickiness and size/excess density characteristics of primary particles, is presented, Fig. 5. Figs. 5(a),(b) show the effect of remineralization on the export flux and its emerged size-spectrum, respectively. In general, the higher the remineralization rate, the less material stays in the system over time to later get involved in the three transformative processes, e.g., aggregation, fragmentation, remineralization. This is directly reflected on the magnitude of the resulted export fluxes, with increasing rem-

ineralization rates producing weaker export fluxes, see annual mean $s-$ratios in Table 1 and the annually integrated export fluxes in the enclosed graph in Fig. 5(b). Regarding the emerged size-spectrum, this analysis suggests that lower remineraliza-tion rates allow the export of aggregates of smaller sizes, Fig. 5(b). The reason for this is connected to fact that small primary particles and bigger but light aggregates stay for longer period in the system, which through their constant involvement in aggregation and fragmentation processes increases their chances to form secondary aggregates of optimal sinking speeds lying

in a wider range of size classes. Lastly, examining the phase diagram, Fig. 5(a), the scenario of $\gamma = 0.05$ produces contradictory results compared to the other two scenarios, with a very rapid response of the export flux to the production of new material. However, the fact that only a very small portion of the latter is exported, while most is lost through remineralization, makes us interpret this result cautiously.

   The stickiness of aggregates affects the export flux in a similar but reversed way compared to remineralization, Figs. 5(c),(d).

Higher stickiness means that once two particles collide, the chances to stick together are higher. The annual mean $s-$ratio increase with increasing stickiness, Table 1, which can be related to the effectiveness of the system to export carbon out of the mixed layer. Regarding the timing that certain events occur in the annual cycle, lower stickiness forces the export fluxes to reach their minimum and start reacting to the production of new material in system earlier in the year (e.g., day 25, 62 and 76 respectively), better visualized in Fig. A1(b,e). Concerning the first point, this might be the case because particles that fail

to form bigger aggregates (in the low stickiness scenario) are forced to stay in the mixed layer where mass is readily removed through remineralization. On the other hand, high stickiness not only supports higher export fluxes during the peak period, but also enables the remaining material to keep repackaging for longer time, in the premise that the strong export event does not strike the system out of material (below a critical level) necessary for the continuation of the aggregation process. Regarding







**Figure 5.** Phase diagram (a,c,e) and density integrated export flux ($f_{100}$) (b,d,f) for a range of different (a,b) remineralization rates $\gamma$ (day$^{-1}$), (c, d) stickiness $\alpha$ (-) and (e,f) size excess density characteristics of the primary particles (S+L refers to small and light, S+D to small and dense and B+D to big and dense). The dots denote the annual mean export flux and production for each scenario. Regrading the density integrated export flux figures, the shadings represent the annual variability, whereas the lines with the circle marks are the annual mean values. The encloses figures show the annual integrated export flux for each scenario.




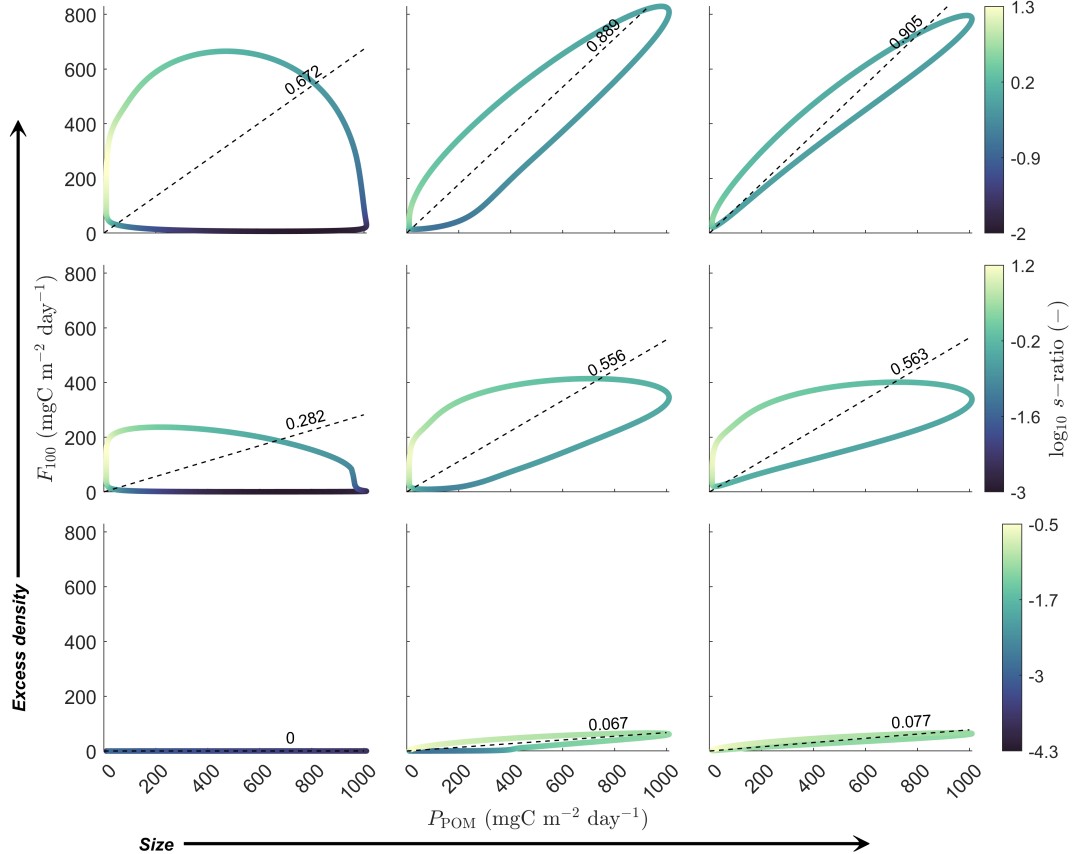

**Figure 6.** Systematic representation of the effect of the size and excess density of the primary particles on the export efficiency of the system, where $F_{100}$ is the total export flux and P is the production of new (primary) particles in the system. The size (radius) range is between 1.27 $\mu$m and 801.4 $\mu$m, where the excess density scaling factor is between 1 and 10. The slope of the dotted lines (see values) indicate the annual mean ratio of $F_{100}$ to $P_{POM}$ ($s-$ratio).

the second part of this observation, even though in the lower stickiness scenario the export flux starts reacting earlier in the year,
it does so in a very low pace, whereas the higher stickiness level system keeps the momentum of the previous year and reaches earlier the critical mass concentration threshold where aggregation becomes the dominant transformative process. Finally, in all cases, the dominant size classes of the resulting size spectrum are around $10^4$ $\mu$m with the difference that higher stickiness means that a wider array of sizes of the exported material are observed, Fig. 5(d), as the stickier the primary particles, the higher the chances that small particles form an aggregate with optimal excess density which enables them to sink fast before
being remineralized.

The size/ excess density characteristics of the primary particles that enter the system also play a crucial role in the development of export flux over time, Figs. 5(e),(f). Our analysis shows that big and dense primary particles (B+D) produce the highest total export fluxes annually, resulting in the highest annual mean $s-$ratio of 0.88, Table 1. Moreover, they do so in the most





**Table 1.** Annual mean $s-$ratios for different remineralization and stickiness values and primary particles characterstics

| Symbol | Description | Value | Annual mean $s-$ratio |
|--------|-------------|-------|----------------------|
| | | 0.01 day$^{-1}$ | 0.92 |
| $\gamma$ | Remineralization rate | 0.03 day$^{-1}$ | 0.32 |
| | | 0.05 day$^{-1}$ | 0.053 |
| | | 0.05 | 0.10 |
| $\alpha$ | Stickiness | 0.10 | 0.32 |
| | | 0.15 | 0.55 |
| | | S+L | 0.01 |
| 2D space | Primary particle characteristics | S+D | 0.55 |
| | | B+D | 0.88 |

responsive way relative to the rate they enter the system. In the case of B+D primary particles, the first aggregates are exported

out of the mixed layer after 50 days, whereas in the case of small and dense (S+D) it takes 141 days before aggregates of optimal sinking speeds are formed, Fig. 5(e). In the latter case, the resulted size-spectrum is skewed to the right with proportionally bigger aggregates being exported compared to the former case, Fig. 5(f). This can be explained by the fact that even though S+D primary particles are denser, their small size means that the aggregation process until the point that big enough for sinking aggregates are formed lasts longer, during which period water is trapped in their interior progressively moving them to lower

excess density and hence lower sinking speed bins. This means that their initial high density advantage is lost through time and they mostly rely on size build-up. On the other hand, a combination of relative lower excess density, but larger primary particles can support a more immediate export flux of variable (and smaller) size-classes. Furthermore, middle-size aggregates that are not dense enough to sink might have more chances to stick with high density primary particles, increasing their excess density dis-proportionally to their increase in size. Finally, small and light (S+L) primary particles result in the weakest export flux

with an estimated $s-$ratio of just 0.01, Table 1, as the biggest proportion of the production is lost through remineralization.

Finally, Fig. 6 presents the effect of primary particles' characteristics, e.g., size and excess density, on the system's dynamics in a systematic way. As one would expect, a combination of larger and heavier primary aggregates leads to the highest export efficiency among all scenarios. In this case, primary particles can either sink fast in their one or form even faster-sinking aggregates in a short time period, which in both cases minimizes the losses due to remineralization. In the other extreme,

small and light primary particles require a lot of time for aggregates of optimal sinking speed to be formed. During this period, remineralization is the dominant transformative process, removing material from the system in a fast rate and hence minimizing its export efficiency. An interesting insight of this analysis is that given a specific size of primary particles, increasing their excess density level has a significant and positive impact on the system's export efficiency. However, given a specific excess density level of primary particles the effect of increasing size is suggested to plateau faster. This figure also showcases how

reactive the export flux is to the incoming primary particles, which is reflected both in the shape of the resulted phase diagram and the range of the instantaneous ratio of export flux to the production of primary particles (see colorbar).





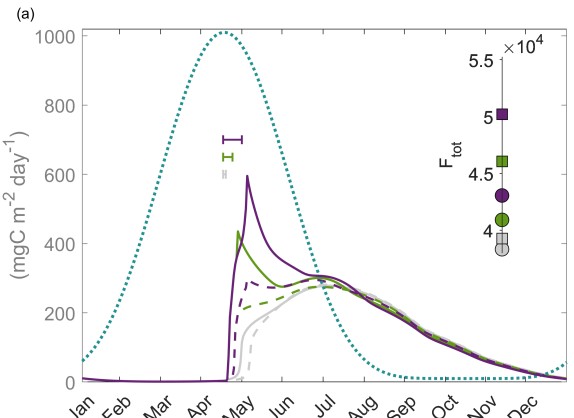
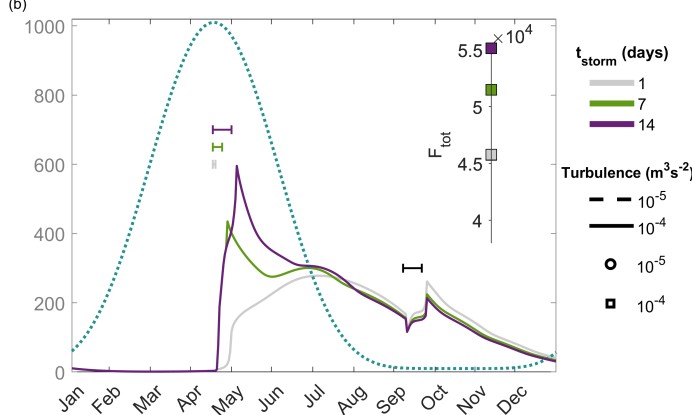

**Figure 7.** Simulation of sudden storm events of variable duration ($t_{storm}$) and intensity (reflected by the turbulent dissipation rate, $\epsilon$). The enclosed graphs show the annual integrated export flux for each scenario. The background turbulent dissipation rate is $\epsilon = 10^{-6}$ m$^3$s$^{-2}$ and the turbulent dissipation rate of the 'fall' storm event in (b) is $\epsilon = 10^{-4}$ m$^3$s$^{-2}$ for all scenarios. The blue-dotted line shows the POM production cycle ($P_{\mathrm{POM}}$) and the colored lines demonstrate the export flux ($F_{100}$) in each scenario.

## 3.3 The effect of turbulence

Turbulence is another important variable and worthy to investigate its effect on the resulted export flux, as it is involved in both the aggregation and fragmentation process. Higher turbulence, on the one hand, increases the encounter rate of two particles (through its involvement in the calculation of the coagulation kernel) and, on the other hand, it makes big, porous aggregates more vulnerable to break down into smaller aggregates of variable excess densities. Sudden and strong storm events have the potential to mix the water column and increase the turbulent levels that sinking particles experience. Fig. 7 attempts to simulate such storms on the same, above-analyzed ecosystem (a simple seasonal cycle where the smallest size-class primary particles with variable excess densities enter a system of a constant background - dissipation rate throughout the year) varying both their duration and intensity. Fig. 7(a) shows that the longer the storm duration is (at the peak of the primary particle production), the stronger the export fluxes that the system can support is, with the effect being more pronounced for higher turbulent dissipation rates in each scenario, and the faster the system's response is. The changes in the total annual export flux (enclosed graph in Fig. 7(a)) indicates the importance of the duration of such storm events, as there is a very small difference between the two storm intensity scenarios in the case of the 1-day storm event (grey lines). The evolution of export flux in time, with a very sharp increase followed by a more gradual decrease, highlights again the strong non-linear relationship between production and export flux. When comparing these scenarios, we can also identify a phase shift around August. Before that point, the longer storm events produce the stronger export fluxes whereas the mirrored pattern is observed (even though it is less pronounced). This might be an indication that these strong episodic events which remove great amounts of material out of the mixed layer might make it difficult for the system to reach the optimal critical mass concentrations needed for aggregation in the following months. On the other hand, a system with a more conservative and steady behavior keeps its material in the system for longer





**Figure 8.** The effect of turbulence on the export flux, as analysed in a highly seasonal (a,c) and a rather constant turbulent environment (b,d). (a,b) present the resulted size-spectrum of export flux (density integrated flux) and (c,d) show the phase diagram with the annual mean export flux and production being denoted in the black dot for each scenario respectively.

time, where even though a part of it is lost through remineralization, the remaining material can more successfully get involved in aggregation processes. This can be better visualized in Fig. 7(b) where during a 7-day storm event in early September the latter scenario responses stronger to the increased turbulence. Moreover, this 7-day storm event leads to an increase of 14.3%, 10.5% and 8.9% in the annual integrated export fluxes (enclosed graphs) in the case of 1-day, 7-day and 14-day early May storm, respectively. An interesting behaviour of the system is observed during the second storm event in Fig. 7(b), where at the beginning of the storm the export flux decreases before rising in a steep slope to reach its peak. This might imply that at






first fragmentation is the dominant process filling the system with smaller-slow sinking aggregates which, in turn, provides the necessary 'fuel' for aggregation to kick off.

In the final part of the analysis, the effect of turbulence in a high seasonal environment with primary particles originating from

an imitated unicellular plankton community is investigated. SISSOMA's output suggests that higher magnitude and variability of turbulence throughout the year leads to higher total export fluxes, i.e., 3.02 E+4 mgC m$^{-2}$ year$^{-1}$ compared to 1.92 E+4 mgC m$^{-2}$ year$^{-1}$ in the scenario of a rather steady and low turbulent dissipation rates, Fig. 8. Hence, the efficiency of the system to export the incoming particles out of the mixed layer varies with an annual mean $s-$ratio of 0.44 in the former case and 0.28 in the later. Higher turbulence might imply that the system possesses a lower critical mass concentration point (where

aggregation boosts the creation of bigger, fast-sinking aggregates) which in turn can be reached faster. This can be observed by comparing the evolution of the export flux in time, Fig. 8(a),(b), where the highly seasonal system supports a stronger flux earlier in the year and hence in lower concentrations of material in the system. On the other hand, the delay observed in the low turbulence scenario leaves primary particles and non-optimal sinking speed aggregates suspended in the water column and exposed to remineralization which decreases the potential of the system to export that material out of the mixed layer. This can

be better visualized in the phase diagram, Figs. 8(c),(d), where during the autumn bloom the export flux in the low turbulence scenario collapses, even though the input of new material in the system is the same for both scenarios.

## 4    Discussion

SISSOMA provides a modeling framework to mechanistically describe the formation and export of aggregates out of the mixed layer. It approaches this by incorporating information about both aggregate size and excess density which are constantly

transformed in this 2-dimensional state space, through three main processes, e.g., aggregation, fragmentation, remineralization, leading to their eventual sinking. The current work focused on how these processes play out over a seasonal cycle and investigated how a variety of parameters, such as remenineralization rate, stickiness, size/ excess density characteristics of the primary particles and turbulence, all affect the intensity and the size structure of the resulted export flux over time. The results of sensitivity analyses underscore the need for more observational and experimental studies on how microbial degradation

acts on marine particles and finding a systematic way to represent how the stickiness of different sourced marine particles varies during the seasonal cycle. In addition, the importance of gaining information about the size/ density characteristics of the primary particles entering the mixed layer has been uncovered.

First, our results highlight the strong nonlinear relationship between the production and the export of material out of the mixed layer, and how these can be described and quantified via the mechanisms involved in transformative processes of aggre-

gate dynamics. Aggregation is sensitive to the concentration of mass in the system, as our analysis has shown there is a clear transition where the formation rate of bigger aggregates accelerates rapidy, a behavior that supports the concept of "critical cell concentration" (Jackson, 1990). These larger, fast-sinking aggregates are are not only exported rapidly but also avoid the high remineralization losses experienced by slower sinking aggregates. This threshold of accelerated aggregation, in turn, is dependent on the size/ excess density characteristics of the primary particles, their stickiness and the prevailing turbulence in



the system. In the first part of the analysis, the observed increase of the export flux is very steep which might appear strange, but it can be explained by the fact that the new-primary particles that enter the mixed layer are assigned to the smallest size-class (1 $\mu$m). In general, the rate of aggregation depends on three encounter kernals, each governed by relative aggregate motion induced by Brownian motion, turbulence and the differential settling respectively, see Appendix A3 . By assuming a constant background turbidity, the aggregation potential of small particles depends on the Brownian motion which means that the aggregation process is very slow, as the smaller the particles the lower the chances to encounter each other (Burd and Jackson, 2009). During this time, material accumulate in the mixed layer at a rate determined by production and remineralization until a point where the concentration is high enough and for the aggregation to really kick off, a process that is very sudden and accelerates rapidly. The story is different when primary particles of variable (and higher) size-classes enter the system, section 3.2, Fig. A2(g,h,i) and Fig. A1(c,f). In this case, both differential settling and turbulent shear contribute to a faster and more efficient encounter of primary particles or proto-aggregates which leads to a more direct response of the export flux to the introduction of new material in the mixed layer. This is also reflected on the instantaneous and annual mean $s-$ratios, Fig. A1(c,f), which are tracking the different export efficiencies between the two systems. The large range of fluctuations of the instantaneous $s-$ratio throughout the year suggests that care should be taken when generalizing the results of short-term field exhibitions, as the measured efficiency of carbon export depends strongly on the sampling time. In this direction, we could even use SIS-SOMA's modeled annual $s-$ratio cycle to apply corrections on the measurements and draw more representative conclusions. Overall, the size/ excess density characteristics of the primary particles directly affect the mass-concentration threshold for the aggregation to become the dominant transformative process which in turn defines the observed time-lag between production and export flux, Fig. 6. As POM is an important energy source for a wide array of organisms, especially in the mesopelagic zone, even small changes in timing might lead to a mismatch between their phenology and the availability of their food with unpredictable cascading effects (Robinson et al., 2010).

Moreover, the smaller the primary particles, the further they are from the optimal for sinking size/ excess density ranges and every time a new aggregate is formed their relative excessive density decreases (due to the inclusion of surrounding water). In a system like this, this might imply that a portion of aggregates might end up becoming big but not dense enough to sink. Once these aggregates reach the size-threshold where fragmentation is dominant, the 2-dimensional state-space fills with aggregates of various sizes and excess densities. This increased diversity of aggregate characteristics combined with the fact that the 2-dimensional state space is filled with material through two directions (new, small primary particles and fragments of aggregates) increases the stochasticity of the system dynamics. On the one hand, it might further boost the aggregation process by redistributing the mass and it might also enable the system to sustain an export flux for longer period after the peak export event, even though a portion of this material is lost in time due to remineralization. On the other extreme scenario of big and dense primary particles entering the system, this effect of fragmentation in time might not be so important, as the newly-entered primary particles can form aggregates which are either rapidly exported or the relative high density of their fragments enables them to repackage and get exported in short time after their re-package. Understanding these mechanisms is very important when we want to predict how shifts in the structure of the plankton community (and hence shifts in the





characteristics of primary particles) will affect the export fluxes in the future, with direct consequences on the global carbon
cycle and the availability of energy sources in the deeper ocean.

Our results indicate that the level of this "critical mass concentration" is also dependent on turbulence and stickiness in
a way that higher turbulence or turbulence leads to a lower critical mass concentration point. This implies that even in the
smallest mass concentrations, particles increase their chances, first, to encounter and, then, successfully stick to each other
moving mass into bigger, fast-sinking size classes. The first part of the analysis in section 3.3 shows how short, episodic events
of high turbulence, such as during wind storms, affect the evolution of the export flux in time and its magnitude. A strong
storm event might at that specific moment enhance the export of material out of the system, however this might lead to a
reduced capability of the system to export material in the following period, as their concentration in the mixed layer stays
lower than the optimal levels needed for aggregation to kick off. Moreover, our results indicate that the duration of these events
define the magnitude of their impact on the export flux, as the system needs a period to gain the momentum and react on
the increased mixing conditions. When comparing a highly seasonally mixed to a stratified system (even though this was a
comparison of two extreme situations and the reality in nature is more complicated) our results could immobilize us to look
closer into the consequences that the increased stratification in deep convection regions, such as the north Atlantic, could have
in their potential to export and sequester carbon. Overall, the seasonally mixed system supports a stronger-bimodal export flux
which comes earlier in the year, whereas the stratified scenario failed to sustain an 'autumn' export event. Apart from all the
above-mentioned dynamics that apply to this analysis too, fragmentation plays its role in a way that depending on what phase
of the annual mixing cycle the aggregate fractures are released in the system, they might have the chance to either rapidly
repackage and be exported out of the mixed layer (in the case of high turbulence), or they will be turned into nutrients in case
the turbulence stays low for long period.

Finally, seeing the full dynamics with all three tranformative processes in play we could identify three characteristic periods,
Fig. 9. First, there is a 'preparation phase' where new primary particles enter the mixed layer, a portion of which might have
the optimal size/ excess density properties to sink fast out of it, but the rest of the material accumulates in the system. By
remineralization being the dominant process, a portion of them is lost into nutrients and progressively moves into lower excess
density bins. This ends when a critical mass concentration level is reached where aggregation becomes the dominant process
moving mass to larger size bins and in intermediate excess density bins compared to the 'parent' particles or proto-aggregates.
The stickiness, turbulence and the relative proportion of size/ density primary particles characteristics all affect the dynamics
of the system and the critical level in which each phase initiates. To better compare the system's variability among the different
scenarios, we introduce a new metric where we follow the level of cumulative new production entering the system until the
half of the maximum export flux is reached, Fig. A4. For example, Fig. A4(a) indicates that higher stickiness not only leads
to higher annual mean export fluxes, but also initiates substantial export fluxes earlier in the year under lower accumulation
levels of new particles in the mixed layer. It is more difficult to draw certain conclusions about the transition between the
export and de-escalation phases. However, according to our understanding higher stickiness, turbulence and proportion of
big/ dense primary particles might lead to a shorter (but more intense) export phase draining the system out of necessary
mass for aggregation to keep being a dominant transformative process. Fragmentation follows the dynamics of aggregation by



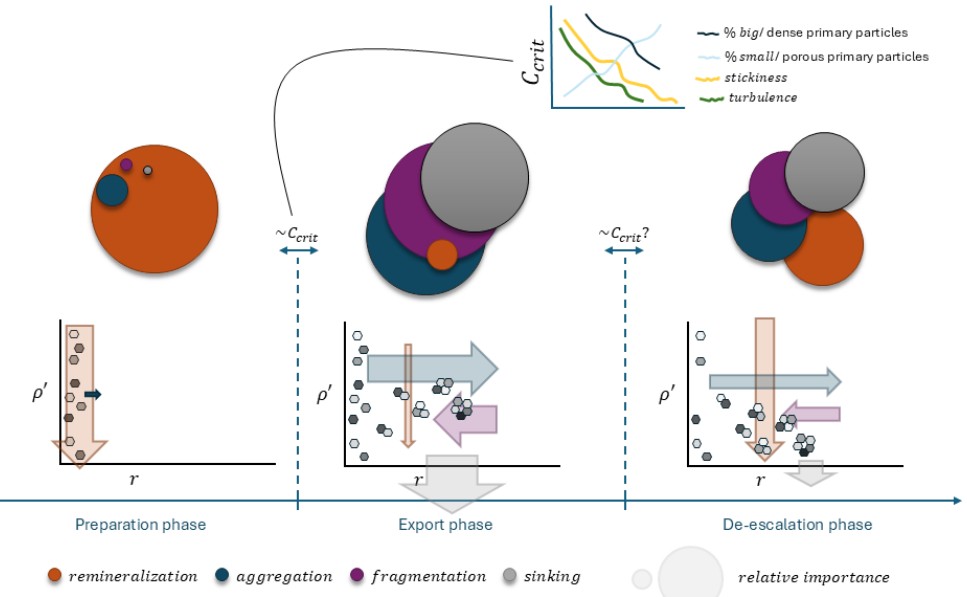

**Figure 9.** Summary of the dynamics of the system in an annual cycle as explored in SISSOMA. Three distinct period can be identified: 1) a preparation phase where remineralization is the dominant tranformative process acting on the primary particles and the small proto-aggregates, 2) the main 'export phase' kicks off whenever the critical mass concentration (which depends on turbulence, stickiness and the size/ excess density characteristics of the primary particles) is optimal. During this period, aggregation, fragmentation and sinking out of the mixed layer are the dominant processes redistributing mass in the 2-dimensional state space and eventual exporting a big part of it, 3) when the system has exported a big portion of the material in the mixed layer (the critical mass concentration falls below a threshold) the 'de-escalation' phase starts where all processes are in play until the cycle starts again.

constantly redistributing mass to smaller size classes of various excess densities. The third and last phase starts when the mass

concentration falls below the critical level. Now, aggregation and fragmentation continue to operate, but remineralization's influence increases steadily. In this way, all three transformative processes are in play, but the system is progressively occupied by lower size and importantly lower excess density material. These slower-sinking aggregates stay now suspended in the water column and most of them are lost due to remineralization, until new production comes and the cycle starts again. The timing of these event and how fast they occur in this cycle takes depends on both the size/ excess density characteristics of the primary

and the environmental conditions, as discussed above.

Even though SISSOMA captures many important factors and processes related to an ecosystem's potential to export material out the mixed layer, there are certain assumptions that need to be taken into account and improved. In the current form, remineralization removes dry mass from an aggregate in a constant rate throughout the year and moves it to lower excess




density bins. SISSOMA can provide a platform for testing a variety of hypotheses on how biological, physical and chemical
factors, as well as the aggregates' properties all affect the degradation rates of the latter. Anderson et al. (2023) suggest that
microbial remineralization roughens the surface area of the sinking aggregates which in turn increases the attachment area for
the microbes to colonize and eventually degrade it in faster rates than expected. In our case, the incorporation of porosity as a
third state variable for the aggregate's characteristics could allow us to incorporate this information when more experimental
data are set to draw a mathematical formulation of this process. Moreover, modeling the effect of ballast minerals on the
sinking velocities and degradation rates is crucial. Dense mineral particles, such as calcium carbonate or opal, usually become
an important part of an aggregate increasing its sinking velocity. This on the one hand shortens the period that this newly-
formed aggregate stays in the upper ocean layers where remineralization is higher and on the other hand there is a contradicting
evidence that ballast minerals can create a protective layer around the aggregate further decreasing the microbial degradation
rates (Cram et al., 2018). It has also been suggested that disproportionally high ballast mineral concentrations might have
totally the opposite overall effect by producing more but smaller aggregates, due to the low proportion of the necessary organic
matter - 'glue' for keeping them together in the long run (De La Rocha et al., 2008). Finally, the way particles are moved
around the 2-dimensional state space during their degradation time can be challenged, too. Contrary to the current direction of
mass, one might argue that by removing the organic-light part of an aggregate, the proportion of ballast-heavy minerals inside
the aggregate increases which in this case might result in heavier, fast-sinking aggregates instead.

Another relevant property to test in the model is the variable lability of the primary particles. It has been proposed that
new production initiated by the input of excess nutrients, mainly in upwelling regions and deep convection areas, is labile
and an easy target for the microbes which turn a big proportion of the sinking material into nutrients. In contrast, highly
recycled material in the tropics and oligotrophic regions are usually refractory (Francois et al., 2002). This might imply that
small, suspended material or bigger but not heavy enough aggregates remain in the system for longer periods increasing their
chances to form aggregates of optimal sinking speeds and support considerable export fluxes in time. In the current SISSOMA
version, the stickiness of the primary particles remains constant throughout the annual cycle. Grønning and Kiørboe (2022)
showed that when certain diatom species sense the presence of copepods in their close proximity, they can quickly increase
their stickiness to form big, fast sinking colonies to escape predation. Moreover, it has been suggested that different phases in
a phytoplankton bloom might produce particles of varying stickiness depending on a wide array of biological, physical and
chemical conditions of the ecosystem (Baumas and Bizic, 2024b). Even though we acknowledge the importance of stickiness
which is also supported by our results, there is a need of further research in the topic to be able to confidently draw the
appropriate mathematical framework and incorporate it in models. The above-described research areas could be approached by
programming separate classes for primary particles coming from different sources, which then the model follows throughout
the simulation. In this way, we could be able to know the proportions of different material in an aggregate, e.g., ballast minerals,
fecal pellets, phytoplankton cells, in any time and decide more efficiently the direction of mass movement.

Last but not least, it has been suggested that the structure and phenology of the plankton community are the main factors
that determine the characteristics of the export flux. In this direction, the next step is to couple SISSOMA with the Nutrient-

Unicellular-Multicellular (NUM) model, a trait-based framework which uses first principles to model the global biogeography of the plankton community (Andersen and Visser, 2023).

## 5 Conclusion

In conclusion, SISSOMA provides a useful tool to mechanistically describe the dynamics of the seasonal cycle of the carbon export flux. This framework helps us to better understand how aggregation, fragmentation and remineralization shape the emerged aggregate community in time, until their eventual export out of the mixed layer. Our results highlight the nonlinear relationship between the production of primary particles in the upper mixed layer and their export as aggregates out of it, which can be reflected on the wide fluctuations of the instantaneous $s-$ratio throughout the annual cycle. Moreover, it has been shown how remineralization rates, stickiness, size/ excess density characteristics of the primary particles and turbulence all affect both the intensity and the size-structure of the export flux. Finally, this study stresses out the need for more, both in situ and experimental, research to be conducted which will help us develop the appropriate mathematical framework and incorporate them into more sophisticated bio-geochemical models.

*Code and data availability.* The code of this model analysis is open source and available on (Kandylas and Visser, 2025) (DOI 10.5281/zenodo.15039330)

*Author contributions.* AWV and AK devised the concept, developed the code and drafted the manuscript. AK implemented and validated the code, and curated the data and code.

*Competing interests.* The contact author has declared that none of the authors has any competing interests.

*Acknowledgements.* This work was supported by the Centre for Ocean Life, a Villum Kann Rasmussen Centre of Excellence supported by the Villum Foundation, by the Simons Foundation (grant 931976), and the European Union's Horizon Europe research and innovation programme under grant agreements 869383 (ECOTIP), 101083922 (Ocean-ICU), 101136480 (SEA-Quester) and 101136748 (BioEcoOcean).



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



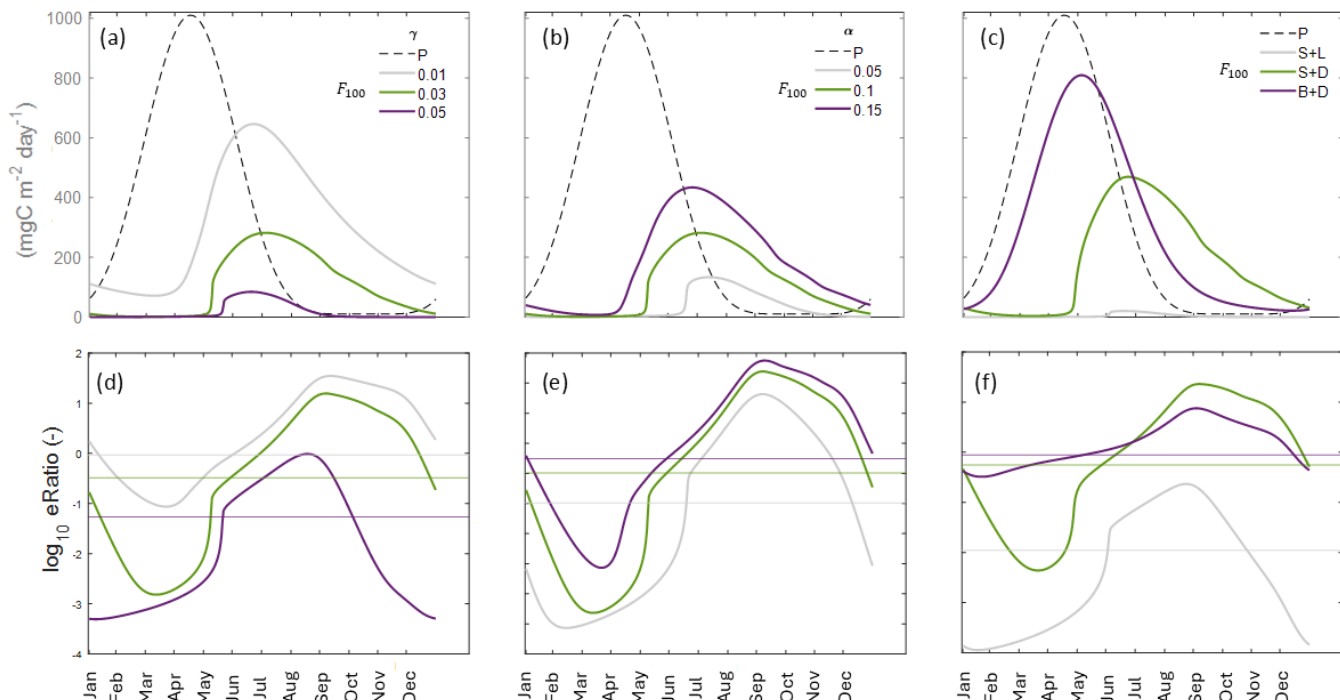

**Figure A1.** Evolution in time of production (P) and export flux ($F_{100}$) (a,b,c) and e-ratio, where the lines indicate the annual mean e-ratio for each scenario (d,e,f) for different values of remineralization (a,d), stickiness (b,e) and size/ excess density characteristics of the primary particles (c,f).







**Figure A2.** Evolution of the density integrated flux ($f_{100}$) in time for different values of remineralization (a,b,c), stickiness (d,e,f) and size/excess density characteristics of the primary particles (g,h,i). Note that the scales of the colorbars vary.



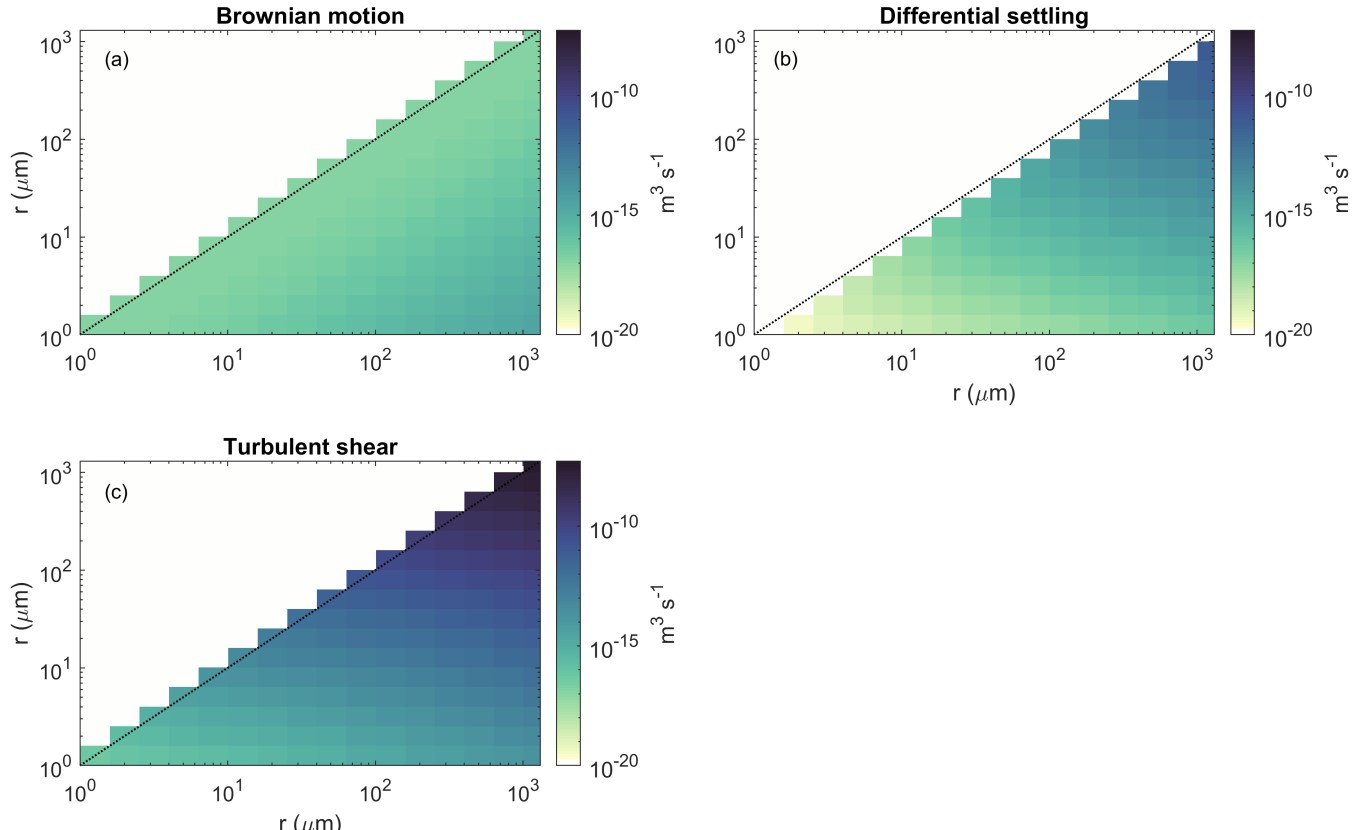

**Figure A3.** The three components of the coagulation kernels: a) brownian motion, b) differential settling, and c) turbulent shear.





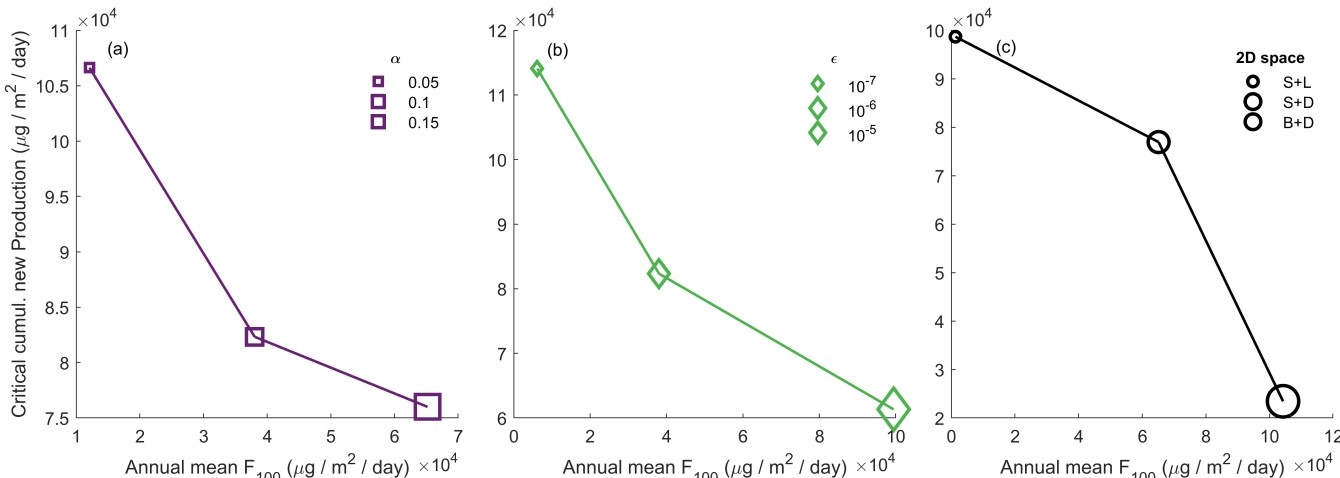

**Figure A4.** Sensitivity analysis of different values of: a) stickiness, b) turbulence, and c) primary particle characteristics. The 'Critical cumulative new production' is a metric that follows the level of cumulative new POM production entering the system until the half of the maximum export flux is reached and it is plotted against the annual mean $F_{100}$.




**Table A1.** Glossary of variables, parameter settings for simulations and units.

| variable | description | Value | Range | Units |
|---|---|---|---|---|
| $a$ | self similarity parameter | 2 | – | – |
| $\alpha$ | stickiness | 0.1 | – | – |
| $\varepsilon$ | Background turbulent dissipation rate | $10^{-6}$ | – | $\text{m}^3\text{s}^{-2}$ |
| $P_{\text{POM}}$ | Total productivity of POM | $10^6$ | – | $\mu\text{gC m}^2\text{ day}^{-1}$ |
| $q_{m:c}$ | Dry mass to carbon mass ratio | 2.5 | | – |
| $h$ | Depth of simulated surface layer | 100 | | m |
| $r$ | Aggregate raduis $r = r_o \delta^x$ | – | $r_o$=1 to $r_{\text{max}}$=$10^6$ | $\mu$m |
| $\rho'$ | Aggregate excess density $\rho' = \rho - \rho_w = \rho_o \lambda^z \delta^{(a-3)x}$ | – | $\rho_o$=1 to $\rho_{\text{max}}$=64.2 | $\text{kg m}^{-3}$ |
| $\rho_w$ | Density of seawater | 1027 | | $\text{kg m}^{-3}$ |
| $p$ | Agregate porosity | – | | – |
| $\phi$ | Aggregate solid mass volume fraction $\phi = 1 - p$ | – | | – |
| $m'$ | Aggregate dry mass $m' = v\rho' + \phi v \rho_w$ | – | | $\mu$g |
| $w$ | Aggregate sinking speed | – | | $\text{m day}^{-1}$ |
| $x$ | Radius bin ordinate | – | $x$=0 to $X$=29 | – |
| $y$ | Density bin ordinate | – | $z$=0 to $Z$=9 | – |
| $\delta$ | Radius logarithmic interval $\delta = (r_{\text{max}}/r_o)^{1/X}$ | 1.58 | | – |
| $\lambda$ | Excess density logarithmic interval $\lambda = (\rho_{\text{max}}/\rho_o)^{1/Y}$ | 1.42 | | – |
| $\mathsf{N}(x,z)$ | Aggregate number density in state space | – | | $\text{m}^{-3}$ |
| $\mathsf{M}(x,z)$ | Total dry mass of aggregates $\mathsf{M} = \mathsf{N}m'$ | – | | $\mu\text{g m}^{-3}$ |
| $\mathsf{F}(x,z)$ | Aggregate sinking flux $\mathsf{F} = \mathsf{M}w/q_{m:c}$ | – | | $\mu\text{gC m}^{-2}\text{ day}^{-1}$ |
| $\mathsf{P}(x,z)$ | Size and density resolved productivity | – | | $\mu\text{gC m}^{-2}\text{ day}^{-1}$ |
| $\psi(x,z)$ | State dependent fragmentation rate | | | $\text{day}^{-1}$ |
| $\beta$ | Coaggulation Encounter kernel | | | $\text{m}^3\text{ day}^{-1}$ |
| $\gamma$ | Remineralization rate | 0.03 | | $\text{day}^{-1}$ |
| $\psi_0$ | Fragmentation rate | 0.5 | | $\text{day}^{-1}$ |
| $g$ | Acceleration due to gravity | 9.8 | | $\text{m s}^{-2}$ |
| $C$ | Drag coefficient | | | $\text{day}^{-1}$ |
| $R$ | Reynolds number | | < 100 | – |
| $\eta$ | Kinematic viscosity of seawater | $10^{-6}$ | | $\text{m}^2\text{s}^{-1}$ |
| $v$ | Aggregate volume, $v = 4\pi r^3/3$ | | | $\mu\text{m}^3$ |