# Peer review of "Seasonal cycles of the carbon export flux in the ocean: Insights from the SISSOMA mechanistic model"

_EGUsphere, 2025_

## Author Comment (AC2)

**Reviewer 1**

This is a mostly competently executed paper, although it has several major issues that make me doubt its publishability in its present form. The English is fairly good, although there are some quirks, detailed below.

The authors have mostly done a good job of separating Methods, Results and Discussion. However, the Discussion meanders and many passages are vague or unnecessary. It could be tightened up. It should also be expanded to include a few extra topics I will detail below, but the length of the current text could be reduced. And the Methods are missing some critical details. Many of the figure captions and legends are inadequate, and one figure is almost incomprehensible (see below Data Presentation).

There are a few references that I was surprised to not see cited (e.g., 10.5670/oceanog.1998.03, 10.1126/science.aay1790, 10.1038/s41598-020-60424-5). I am not suggesting that the authors 'shoehorn' these in if their inclusion borders on gratuitous, but I would recommend that they at least read the papers and think about whether there might be a place for them in the revised Introduction or Discussion.

**Major Issues:**

(1) I'm not sure what the overall purpose is. The paper doesn't really break any new ground with regard to the theory of particle aggregation. Nor does it present a novel computational tool to the modelling community in a way that seems likely to encourage its widespread adoption. It takes a tool that the authors have developed, conducts some numerical experiments with fairly idealized hypothetical environments, and reiterates some known facts about nonlinearity of aggregation response to particle concentration, sensitivity to stickiness, etc. If these authors really want the community to adopt this tool, one would think they would (a) do some evaluation of it relative to the available alternatives, (b) exhaustively document what its required inputs are, and (c) include some discussion of the computational cost, which I would assume is high compared to the (admittedly simplistic) formulations normally used in global and regional ocean biogeochemical models. There is also concern about traceability, as the antecedent manuscript by Visser et al is listed as just authors + title, with no information about where it is published or submitted (528). It is cited at least twice in the text as "Visser et al" (no year given) (70, 78). Possibly this is because it is submitted but not yet published and the authors expect these details to be available by final publication of this paper, but the details still need to be specified.

We thank the reviewer for his careful reading of the manuscript and constructive comments. We do not agree with the reviewer that the manuscript breaks no new ground. True, the theory of particle aggregation and sinking has been around for over 100 years, while their fractal nature has been a useful conceptual feature used in modelling descriptions for decades. What has not been done before is (1) casting this description into a multidimensional state space (2) including explicitly remineralization and fragmentation and (3) abandoning the fractal dimension as a descriptor of aggregate morphology to self-similarity as a descriptor of the aggregation process. The latter is quite subtle but core to the model framework, and addresses a key issue; that under remineralization and fragmentation, fractal dimension can no longer be

considered a conserved aggregate property. We doubt the reviewer has ever encountered the full state space formulation in equation (4) before. Neither will he have seen the convolution algorithm that provides the engine for simulating aggregation – remineralization – fragmentation processes. The usual approach is to either use Monte Carlo simulations, or to rely on some fractal dimension or particle size spectrum simplification.

Further, we do not intend for SISSOMA to become an operational model adopted broadly across the oceanography community as seems to be inferred from the reviewer's comments. We hope that the reviewer is aware that models can serve many purposes of which operational application is one. Models also serve as research tools that can highlight the macroscopic effects of complex processes. With this in mind we seek to address a specific issue, namely the relationship between the production of particulate organic matter (POM) and its export into the ocean interior; the s-ratio factor of the elusive e-ratio that is such a contention issue the ocean carbon cycle (equation 1). Existing models for export (e.g. Seigel 2016, Laurenceau-Cornec 2023) used on a global scale do not consider aggregation per se, but rely on empirically derived constant sinking speeds of different classes of material. The seasonal cycle of POM production provides an important illustration of the processes involved. Yes there are nonlinearities associated with concentration, and stickiness is also important, but so too are remineralization and fragmentation, with the relative influence changing over time. Hence the extensive use of phase plots linking POM production to export flux, where we seek to highlight the role of aggregation-remineralization-fragmentation on seasonal export cycles. These variations are almost completely ignored in the state-of-the-art e-ratio models mentioned above. We hope that this is now clear in the introduction, and that the reviewer can see the rationale for this study.

We do not seek to resolve the full e-ratio description in this manuscript as this depends also on the structure and succession of the plankton community. This is the next step in the puzzle. Here we can also note that SISSOMA is ideally structured to integrate with trait-based plankton models such as NUM that resolve size structured phyto- and zooplankton seasonal cycles.

The Conclusion begins "In conclusion, SISSOMA provides a useful tool to mechanistically describe the dynamics of the seasonal cycle of the carbon export flux." But is this really a conclusion, or an a priori assumption? I'm not saying that the tool is not useful, but as a statement of what was actually demonstrated by the data shown in this paper, this doesn't quite work.

It is true that the model is built on already established knowledge, but in our knowledge, there is no other framework putting together all these together. The overall purpose of this paper is to demonstrate that SISSOMA is an easy-to-use tool which can be used to test a variety of hypotheses about export fluxes.

Added sentence in the Conclusion: 'Although the theory and implementation of the aggregation processes are well established, more work needs to be done to improve the function of the fragmentation and remineralization processes in the model. This stresses the need for more, both in situ and experimental, research to be conducted, which will help us develop the appropriate mathematical framework and incorporate them into more sophisticated biogeochemical models.'

The sampling data following a full seasonal cycle is yet scarce in the scientific community to allow us to extensively validate our result. Lack of similar projects to compare with.

The required inputs are presented in table A1, and we made sure that we refer early to it in the methodology part (first paragraph of the Methodology).

Regarding the computational cost, SISSOMA, in its current form, is more suitable for understanding the export flux dynamics on a regional scale. (end of the first paragraph in Discussion)

The analysis in sections 3.1 and 3.2 are based now in a different seasonal cycle, see Fig. 2.

The "Visser et al" paper was a pre-print and now it is removed from our references

(2) To follow up on point (b), the documentation is inadequate. My first reaction to equation (3) was that there are multiple symbols that are never defined.

We assume the reviewer is referring to equation (4) not (3) as the symbols in (3) are all defined in the adjacent text.

Actually most of them are defined in Table A.1, but there is no reference to this Table until Section 2.2. I count three that are not, but are defined in the text on 103-111. None of these descriptions are very specific or informative.

We are not sure what it is the reviewer finds lacking in the description. We updated Table A1.

Possibly this paper is just an application of what is in the cited literature and contains no novel process parameterizations. But again, I think the authors need to decide what their objective is. If this paper is presenting a novel process model, there needs to be a much more detailed description of the model itself. If it is just an application of existing model to present a new tool to the community, there needs to be more emphasis on the computational framework and potential future applications.

The authors do not always clearly define their terms (e.g., f100 vs F100) and some critical details about their idealized hypothetical environments are not explained. When f100 and F100 are introduced (equations 4 and 5) the meaning is fairly clear, except that x and y are not defined (see below Figure 1). As I read this, f100 is the export from a layer that we will treat for now as 100 m thick, integrated over the spectrum of excess density (rhoprime), in each increment of size (dr), and F100 is the flux integrated over both r and rhoprime. This is mostly consistent with the remaining text and figures, except for 166 where F100 is described as the "density-integrated export flux" which is defined as f100 on 144-145. On 178-180 there are several references to "the total amount of exported material, Fig. 4(c)". But Figure 4c actually shows f100; it's fairly clear from the plot that the statement is correct, but if one is going to define these quantities, then the text should reflect those definitions.

On 140 the vertical extent of domain of interest is stated to be "100m", which is consistent with the choice to call export flux F\_100. On 150 we have "h is the mixed layer depth", implying that it is variable, which appears not to be the case. In Table A.1 it is described as the "Depth of simulated surface layer" and a constant value of 100 m is specified. In Figure 3(c), only a single value of TKED(t) is given, although the caption could lead readers to believe that h is not constant ("two scenarios were used to represent the stratification conditions of the system: a highly seasonally mixed and a stratified throughout the year.") The Abstract states that "The effect of increased stratification ... is then presented and discussed" and yet the experimental design appears to be built around a uniform mixed layer depth in all cases.

I would like the authors to (a) make a clear up front statement of what their idealized vertical structure is, e.g., a mixed layer with a constant depth of 100 m, that is vertically uniform and well mixed with a TKED that is constant in depth but variable in time, (b) add a paragraph to the Discussion that explains that this is a highly idealized case, and that in the real world the layer thickness would covary with the TKED, and the rate of particle production would vary with the flux of nutrients brought into the layer by mixing and entrainment.

Also I'm not sure the 1/h in eqs 4 and 5 is necessary. If we have a well-mixed layer of thickness h with a concentration of particles X with an average sinking rate w, then dX/dt=wX/h (see eq. 3) is the rate of change within the mixed layer (mg m^-3 d^-1), not the flux across its base (mg m^-2 d^-1).

- Improved table A1 in Appendix
- The required inputs are presented in table A1, and we made sure that we refer early to it in the methodology part (first paragraph of the Methodology).
- All symbols of the equation (3) are described in lines 102 to 114.
- Added definitions of the logarithmic scaling factors (x,y)
- Fixed line 166 (f100)
- Removed "the total amount of exported material, Fig. 4(c)" (line 178)
- Good point about the mixed layer depth. Even though we use the term 'stratification', we keep the mixed layer depth constant at 100m. The 'stratification' scenario changes only the turbulent dissipation rate.
  - Added paragraph: 'It is important to mention that, in the context of this project, in the seasonally stratified scenario only the turbulent dissipation rates varies with time, while the mixed layer depth remains constant at 100m throughout the annual cycle in all simulations.' (line 145 of revision)
  - Added new paragraph in the discussion about the seasonal variability of the mixed layer depth (line 404-411 of revision)
  - o Added clarification in the caption of figure 3
- It is correct as it stands, the unit w/h has units day-1, which rate of change of concentration in the mixed layer.

(3) When I look at the results regarding sensitivity to remineralization rate (Table 1), I have doubts about the credibility of this tool. The s-ratio varies almost from 0-1 over a very narrow, and somewhat implausible, range of 0.01-0.05 d^-1. I think values 10X as large would be more realistic and more representative of what is used in contemporary ocean models, yet in this model those values would all be in the region where the response is asymptotic and sensitivity negligible (Figure 5). This should certainly be discussed in the Discussion.

The previous analysis was using primary particles of the smallest size class, 1  $\mu$ m, which was an extreme scenario. In the new analysis we use primary particles in the range of 1-50  $\mu$ m and remineralization rates 0.2, 0.6 and 0.1 (1/day).

There are studies recommending  $\gamma$  = 0.1 day–1 (Kiørboe, 2001; Cavan and Boyd, 2018; Bach et al., 2019). However, other studies with direct observations suggest a lower value ( $\gamma$  < 0.03) (Belcher et al., 2016).

Kiørboe, T.: Formation and fate of marine snow: small-scale processes with large-scale implications, Sci Mar, 65, 57–71, 2001.540 Kranenburg, C.: The Fractal Structure of Cohesive Sediment Aggregates, Estuarine, Coastal and Shelf Science, 39, 451–460, https://doi.org/https://doi.org/10.1006/ecss.1994.1075, 1994.

Cavan, E. L. and Boyd, P. W.: Effect of anthropogenic warming on microbial respiration and particulate organic carbon export rates in the sub-Antarctic Southern Ocean, Aquatic Microbial Ecology, 82, 111–127, 2018.

Bach, L. T., Stange, P., Taucher, J., Achterberg, E. P., Algueró-Muñiz, M., Horn, H., Esposito, M., and Riebesell, U.: The influence of plankton community structure on sinking velocity and remineralization rate of marine aggregates, Global Biogeochemical Cycles, 33, 971–994, https://doi.org/10.1029/2019gb006256, 2019.

Belcher, A., Iversen, M., Giering, S., Riou, V., Henson, S. A., Berline, L., Guilloux, L., and Sanders, R.: Depth-resolved particle-associated microbial respiration in the northeast Atlantic, Biogeosciences, 13, 4927–4943, https://doi.org/10.5194/bg-13-4927-2016, 2016

In Figure 4a daily (?) data are shown and the s-ratio takes values > 15 (193). I think it would be better to only cite in the text e.g., monthly averages, and include some Discussion of whether these high values are realistic and how they compare to observation-based estimates of this ratio. There appears to be a rather lengthy period in the fall where F\_100 >> P\_POM, so even in a monthly mean s>>1. But could this also be related to the low remineralization rate? Is it really plausible that below a critical threshold for aggregation, organic particles will sit in the mixed layer for months and not decompose?

These high values of s-ratios are a result of the delay effect between production of material and export. It takes time for very small primary particles (which was the case in that analysis) to form fast-sinking aggregates, and this means that while the production is big the export can be low and vice versa.

(4) It should be acknowledged somewhere that Stokes' Law is not an accurate description of the relationship between size and sinking rate in natural marine or lacustrine aggregates, and this has been known for a long time (e.g., Hawley, 1982, JGR 87: 9489). Also the 'modified' Stokes' Law (93-96) needs further explanation. No reference is given, although possibly it is explained in the currently untraceable reference by Visser et al (20\*\*). The basic formulation of Stokes' Law does not include a drag coefficient, so some explanation of what this represents and how its value was derived is warranted. No value is given in Table A.1. Its unit is given as d^-1 (a drag coefficient for wind stress at the ocean surface is nondimensional).

We added information about Reynolds number and drag coefficient in table A1.

**Terminology:**

I would recommend to remove most or all instances of "chance" and "chances". Some may be innocuous, although unnecessary (e.g. 224). Others (e.g., 110, 187, 204) create a misleading impression that there is a stochastic component to the model (same goes for "probability" on 109).

**Ok, removed but one.**

Inexperienced authors often overuse words like "dynamics". I count 26 in this MS. It can be a useful exercise to go through the MS searching out each occurrence of this word and at each instance ask (a) would another word serve as well, or make the meaning clearer? and (b) would the MS lose anything important if the word were not used at all? This is one of those scientific words that has specific meanings in specific contexts, but can also serve as an all-purpose buzz-word. To rethink each usage as suggested will help the author to develop his scientific writing style. The caption to Figure 4 begins with "system dynamics" and panel (d) is described as "the system-characteristic phase diagram"; this doesn't really help the reader to understand what is being shown.

Reduced the use of the word to three instances.

On 335-337 these two concepts (stochasticity and dynamics) come together in a statement that "increased diversity of aggregate characteristics ... increases the stochasticity of the system dynamics". This is vague, unnecessary and bordering on meaningless. Puff-phrases like this should be avoided. And again: there is no stochastic component to this model.

**Removed**

I would also review all occurrences of "optimal". I don't think it's really clear what an 'optimal' sinking speed (e.g., 188, 204) or an 'optimal' excess density (e.g., 224) means in this context. Export is a monotonic function of sinking speed and sinking speed is a monotonic function of density.

**Fixed**

"phase" is another nonlinear dynamics word that can be overused (20 occurrences). Possibly in this paper it is necessary (e.g., Figures 4 and 5). But there are also cases where it is clearly unnecessary, e.g. on 266, "we can also identify a phase shift around August". "bimodal" is also probably unnecessary (358)

Fixed. We use the word phase for 'phase diagram' and describe the three stages in the summary figure 9.

Alongside some of the unnecessary jargon, there are also some oddly colloquial (or teleological) expressions, like a process "kicks off" (177, 278, 317, 353, F9 caption), "the remaining material can more successfully get involved in aggregation processes" (271), "the relative high density of their fragments enables them to repackage and get exported" (342), "to encounter and, then, successfully stick to each other" (348), or "the system needs a period to gain the momentum and react" (354) that seem out of place in this kind of publication (this is not an exhaustive list). Again, a good exercise would be to examine each instance and ask whether such terminology is necessary and whether it makes the meaning clearer for the reader.

**Fixed**

**Data Presentation**

Figure 1 - The description given of the x and y axes ("x and y are the scaling factors") is vague; the term "scaling factors" is not defined nor used elsewhere. On 102 we have "N(s) is the number of aggregates in a given state-space bin, s = (r, rhoprime)". So we might assume that x and y are r and rhoprime. But the why would rhoprime be both one of the variables displayed and one of the axes? Also the colours named in the caption (green and purple), don't really describe the colours shown. (x and y are defined in Table A.1 as the indices to the bins of r and rhoprime but there is no reference to the Table in the figure caption.)

- Added description of x and y in Methodology
- Added description of x and y in caption and reference to equation 2, for readers to read more if needed
- Added 'The colors of the dashed boxes in (a) ..' to more clearly point to the boxes

Figures 2 + 3 (panel (b) in both cases): Axes again labelled generically as x and y, again hints that these are equivalent to r and rhoprime, but not stated explicitly. In Figure 2 it is clear from the

caption which is which (x is r and y is rhoprime); in Figure 3, I think that the same convention is followed, but it's not entirely clear from the caption.

- Added description of x and y in caption and reference to equation 2, for readers to read more if needed
- Added clarification about the 'stratification'

Figure 3 - I think the units of TKED are m^2 s^-3 not m^3 s^-1 (https://cfd-online.com/Wiki/Turbulence\_dissipation\_rate) (see also Figure 7). Also it's not clear what "equally distributed" means in this context.

**Fixed in the figure and text**

Figure 4 The caption is generally good in terms of explaining what is shown, except that the black dot in Panel (d) is not explained (this I the first plot where this device I used; it is explained e.g. in Figs. 5+8). Also in purely aesthetic terms, if you have 4 panels, why not arrange them in a 2x2 matrix? And why not use the same font/colour in the RH axis in panel (a) as in panels (b-d)?

- Added explanation of the black dot
- The idea is that a, b and c panels have time in their x-axis and so it is easier to follow in parallel the progression of P\_POM, F\_100, f\_100 and m\_dry throughout the seasonal cycle
- Changed the color in the RH axis in panel (a) to black

Figures 5 and 7 - I think the inset or "encloses" (sic) plots need some more explanation. As I read it, their position relative to the x and y axes is arbitrary, because they have their own y axis and because Ftot is an integral over dr. But I think this need to be stated explicitly and the units stated in the caption.

Added in caption 'The encloses figures show the annual integrated export flux, F100 (mgC m-2 year-1), for each scenario.'

Figure 6: The caption again begins with a vague, mostly meaningless phrase ("systematic representation") and then mostly fails to explain what is shown. Sorry but this is the one figure that left me mystified. The panels seem to represent increments of r and rhoprime, but only a range is specified in the caption. If 3 discrete increments were tested, why not just state what they are?

Added in caption the three size and excess density values used

And if  $s = F_100/P_POM$ , then a straight line should represent a constant value. But if s is also the colour scale, how can the points where the loops intersect the lines have such similar colours when the slopes of the lines are so different?

- The straight line has a slope equal to the annual mean s-ratio (F100/P\_POM) which is stated in the caption
- In this case, the interception of the dashed line with the phase diagram does not say anything in particular.

What do the loops represent? Daily data over the course of a year? How does the reader tell where Jan. 1 is and which direction the temporal progression goes (anticlockwise, as in Figure 5?)?

 Added red dot denoting the start of the annual cycle and pointed out in caption that it progresses anti-clockwise

Qualitatively, I think key message of the plot is fairly clear. But in terms of proper documentation it fails. The text accompanying this figure (241-251) is also vague in places; it has a bit of an "arm-waving" feel to it and doesn't help the reader much in terms of understanding what the figure tells us about the underlying physical processes.

Added sentence at the end of the paragraph: 'For example, focusing on the case of the highest excess density of 63 kg m-3 but the smallest primary particles, there is a great mismatch between production and export with the highest s-ratio values recorded at the end of the production cycle.'

**Figure 7**

- Panel (b) is mentioned in the caption, but panel (a) is not.

**Fixed**

As in Figure 6, the reader can make an educated guess at what is being shown, but it should be spelled out more explicitly.

**Reorganized the caption for better flow**

As in Figure 5, the x/y position of the inset figures should be explained and the units of their axes specified.

**Added description and units in the caption**

As in Figure 3, the units of TKED are incorrect.

**Fixed**

Also the horizontal bar that indicates storm duration has no colour code in the fall case. Possibly it's not necessary as only one duration was considered. But the text seems to be saying it was 7 days (272-275), the length of the bar more resembles the 14 day case, and as there is no colour code it's hard to be sure.

**Fixed in the text (it should be 14 days)**

Figure 8 - The seasonal cycle of TKED that is used in these experiments needs to be stated explicitly, and maybe shown in a Supplementary figure.

Improved the caption with information about the experimental setup and referring to the methodology section and figure 3 for further information

**Figure 9** - Again leads with an unnecessary jargon phrase "Summary of the dynamics of the system". This cartoon is actually quite clever, and with a bit of attention to detail it could be useful.

**Rephrased**

But as in the previous examples, the authors do not pay enough attention to making sure that all of the symbols and axes are defined.

Mostly importantly, the meaning of the dimensions of the arrows is not stated. Is it possible to make all of the arrows the same width, or the same length? Is it really necessary to have them vary in two separate dimensions, and how can the meaning of these dimensions be communicated to the reader?

- Changed arrows. Same length and their relative importance is denoted by their width.
- Made it clear in the caption

The inset plot at the top has no x axis label. It is possible to guess at its meaning but it would be better if it were explicitly stated in the caption.

**Fixed in the caption**

Also the colours of the arrows don't exactly match those in the legend, although it is fairly clear which is which.

The arrows have increased transparency on purpose, so they animated aggregates are visible too. I decreased their transparency a bit.

Why does the legend on the inset plot have "big/dense" and "small/porous" primary particles? Generally we think of larger particles as having greater porosity, although that is more for aggregates. I'm not sure porosity is even a relevant property of primary particles, or whether why this is so is explained elsewhere.

Changed porous to light. The porous was indirectly referring to the density, but I agree that it is confusing

In the caption, please change "is optimal" to "exceeds the threshold level" or something similar.

**Changed**

**Some details:**

11 and elsewhere change "resulted" to "resulting" (multiple occurrences) Fixed

18 add "in the deep ocean" after "stored" Fixed

20-21 "export is ultimately governed by primary productivity". I don't think this is true; there is a positive correlation across regions, but export ratios are highly variable.

It is true that there are combinations of high productivity – high export, high productivity – low export or low productivity – high export systems, which depend on a variety of factors. This statement is general in the sense that we need production to have export in the first place and through this project we aim to understand what determines the above-mentioned combinations.

28 at this point, P\_POM has not yet been defined; also this would be a good place to state what subgroups of particles they envision as primary POM: does it include living cells? all kinds of cells, or only some kinds? what about colloids (gels) formed abiotically from DOM? or other materials like aeolian mineral dust?

- it is described directly after it is mentioned: '... PNPP and PPOM, the production rate of primary particulate organic material'
- we mention the possible kinds of material in line 80 (of the revised version): 'The aggregation model can use as input a description of the formation of primary particles, representing for instance dead and dying plankton cells, fecal pellets, and/or aolean dust deposits which are all characterized by their size and excess density.'
- We discuss stickiness more in depth in the discussion (section 414-429 of the revised version)

54-57 This strikes me as an overgeneralization. In the Pacific, the mid-latitudes include both regions that have strong seasonal convection and regions that do not. What does "dominated by the microbial loop" mean?

**Removed**

60-61 a very general statement; vague and unnecessary

Even though it is a general statement, we think that this sentence includes all the elements that we are interested in and looking into later on the paper.

82 "aeolian" misspelled Fixed

98 reference format inconsistent (cite/citep) (see also 151) Fixed

104 q\_m:c is not a term, according to the usual definition (e.g., if Z = aX+bY, the terms in the equation are aX and bY) Fixed

111 "losses" misspelled Fixed

112 delete "that it" Fixed

131 stray 'before "spring" Fixed

133-134 change "diffuse sparse nutrients into their cells more efficiently" to "take up nutrients more efficiently at low concentrations" Fixed

134 change "in lower magnitude" to "of lesser magnitude" Fixed

135 add "it" after "dies" Fixed

137 "base" should be "case"? Fixed

154-161 don't think this paragraph belongs in the Methods. Introduction or Discussion.

**Removed**

166-167 "as well as it illustrates the instantaneous ratio between the export flux to the production of new particles" and illustrates the instantaneous ratio between the export flux and the production of new particles Fixed

175 delete "observed" Fixed

177 "there is a critical concentration of mass" yes, but could there not also potentially be a threshold in (r, rhoprime) space?

179 3 significant figures should suffice (see also 193)

184 "keep supporting the formation of optimal-sinking velocities aggregates" I can't tell what this means. Fixed

188 "remineralization" misspelled Fixed

197, 201, 428 change "emerged" to "emergent" Fixed

F5 caption "regarding" misspelled; "enclosed" misspelled Fixed

223 "higher stickiness means that a wider array of sizes of the exported material are observed, Fig. 5(d)" Is this really true? If you draw a straight line across the distribution at a given f100, you will get a broader range with the higher peak. But maybe this is an illusion: the distributions look to me like something close to linear multiples of each other. Higher stickiness means more export, but I'm not sure you can conclude from these data that the distributions meaningfully differ.

**Removed**

233 "until the point that big enough for sinking aggregates are formed" until the point where aggregates large enough to sink form Fixed

234 "during which period water is trapped in their interior progressively moving them to lower excess density" during which period porosity increases and density decreases Fixed

239 change "dis-proportionally" to "disproportionately" Fixed

242 "primary aggregates" should be "primary particles"? Fixed

255 delete "(through its involvement in the calculation of the coagulation kernel)" Fixed

256 change "excess densities" to "excess density" Fixed

258 change "the same, above-analyzed ecosystem" to "the same ecosystem discussed above" Fixed

262 "the faster the system's response is" I can't tell what this means; please try to refer to specific facts that the reader can verify from the data shown. Fixed

267 change "mirrored" to "opposite" Fixed

273 "the latter scenario responses stronger"; change "responses stronger" to "responds more strongly", and try to make clear to the reader what "latter scenario" is being referenced (unclear antecedent) Fixed

279 change "high" to "highly" Fixed

281 2 significant figures is enough

283 "the incoming particles" incoming from where? newly produced particles? there should not be any external sources in this model Fixed

284 change "later" to "latter" Fixed

297 "remineralization" misspelled Fixed

314 change "turbidity" to "turbulence" Fixed

323 change "exhibitions" to "expeditions" Fixed

347 delete "or turbulence" Fixed

356 not sure what "immobilize" means in this context; possibly the intended word was "motivate" Fixed

361 change "fractures" to "fragments" Fixed

362 change "turned into nutrients" to "remineralized" Fixed

366-368 "By remineralization being the dominant process, a portion of them is lost into nutrients and progressively moves into lower excess density bins." Not clear what is the relationship between remineralization and excess density: remineralization is not a function of particle size or density, so it should not directly affect the distribution of density. There's something else they are trying to express here and I can't tell what it is.

- As described in Methodology, we assume that remineralization in SISSOMA removes mass from an aggregate without changing its size. In this way, an aggregate of certain excess density is progressively moved to lower excess density bins in the 2D state space.

'The next term describes loses from remineralization, which we assume occurs at a constant rate throughout the state space and removes only dry mass leading to aggregates of the same size-class but reduced excess density'

432 delete "out" Fixed

519 the doi given does not match the title cited; possibly these authors read an earlier version Fixed

---

## Author Comment (AC3)

**Reviewer 2**

**Review of `Seasonal cycles of the carbon export flux in the ocean: Insights from the SISSOMA mechanistic model` by Kandylas and Visser**

Kandylas and Visser present a study using SISOMA, a model previously published as a preprint by Visser et al. (2024), to explore the seasonal cycle of carbon export flux in the ocean. The model simulates marine aggregates, including their size and excess density, resulting in a time-varying sinking speed. The authors conduct several sensitivity analyses related to particle stickiness, remineralization, and size—excess density characteristics. While the main focus is on the seasonal cycle of the carbon flux, the study also analyzes the s-ratio and its relationship to parameters.

I found the paper well-organized and relevant to important research areas concerning particulate organic carbon and its fate in the ocean. I particularly appreciated the authors' discussion (could be expanded) on how this type of modeling framework can be implemented in more complex ocean biogeochemical models, which are commonly used to study metrics such as the e-ratio and s-ratio.

The manuscript is well written and follows a logical structure with clearly defined sections: Introduction, Methods, Results, and Discussion. It is generally understandable and well presented. However, some points should be addressed or corrected before being considered for a publication. These are outlined in detail below.

I have one major comment: the SISOMA (v1) model was already published as a preprint by the same authors in November 2024. It would be helpful if the manuscript clearly states whether this study is an application of the previously developed tool or if there are significant updates or developments to the model specific to this study. This clarification is necessary, especially since some parts of the manuscript (e.g., Eq. 3 and model descriptions) appear to be very similar to the preprint.

First, we would like to thank the reviewers for their useful feedback and time spent into reviewing this manuscript.

The "Visser et al" paper was a pre-print and now it is removed from our references. For this reason, we expanded the methodology part of this manuscript.

The analysis in sections 3.1 and 3.2 are based now in a different seasonal cycle, see Fig. 2.

**Introduction**

Line 28: "PPoM "is not introduced before it is used.

Fixed. 'the first being the relationship between PNPP and PPOM which is the production rate of primary particulate organic material'

• Line 35: I suggest moving this equation to the Methods section. It can also be given as a written description. Since it is one of the equations used in the analysis, presenting it in the Methods would be more appropriate.

We would like this equation to be in Introduction since the whole manuscript is built around the idea about distinguishing e-ratio from s-ratio.

- Line 44: A parenthesis is needed before "e.g." Fixed
- Line 70: (Visser et al.,) the year is missing, and the DOI is also missing from the references. I believe it is critical to cite this reference correctly, as the model used in this application was described in 2024. The same issue appears elsewhere in the manuscript and should be corrected throughout (I won't point them all out individually).

The "Visser et al" paper was a pre-print and now it is removed from our references. For this reason, we expanded the methodology part of this manuscript.

**Methods**

• **Line 87:** The statement starting with "In principle, ..." belongs more in the Discussion section. In the Methods, the focus should be on describing what has actually been done.

**Fixed**

• Line 101 – Eq. 3: This equation is the same as Eq. 8 in Visser et al. (2024), where it is explained more clearly. I'm not suggesting it needs to be copied word by word, but the explanation could be improved in this version by referring to that paper (depending, of course, on the publication status of the preprint).

We expanded the methodology part, improving the flow and describing all relevant parameters

• **Figure 1:** It would help the reader if the axis labels indicated whether x and y are scaling factors.

Added better description of x, y in the figure captions and methodology

• **Line 144:** "Throughout the report" — I think you mean manuscript. I suggest using `manuscript` instead of `report.`

Changed to 'Throughout this analysis'

In the same paragraph and following equations, there are some inconsistencies. For example, it's unclear what a, b, c refer to — please use equation numbers for clarity.
 Also, Eq. 6 is the s-ratio, but in the text, it is described as Ftot. These should be carefully checked.

**Fixed**

• **Lines 155–160:** This section seems more appropriate for the Introduction rather than the Methods.

**Removed**

**Results**

• Line 163: The subheading "Model Mechanics" is clear for a modeller, but for a broader audience, it would be helpful to provide a more descriptive title that reflects the narrative of Figures 4 and 5.

**Changed to 'Model analysis'**

• **Figure 4:** The blue line in panel (a) needs a legend. Additionally, the meaning of the black dot in Figure 4d should be included in the figure caption — I assume it represents the annual mean, as indicated in the caption of Figure 5.

Added description of the black dot

We don't use a legend for the blue line, because it matches the color of the left x-axis. However, we state it now in the legend for clarity.

• Figure 6: In the caption, I assume P stands for PPOM.

**Fixed**

• Line 226–227: The reference to Figure 5 should be placed in parentheses.

To be consistent with the rest of the manuscript, we refer to the figures without using parentheses. We changed it to Figs. 5(e,f).

**Discussion**

- Line 293: "SISOMA provides modeling... mixed layer" but throughout the study, 100 m depth is used. Since the mixed layer depth varies seasonally, was this variability accounted for in your application? I think it is important to be consistent and clear.
  - Even though we use the term 'stratification', we keep the mixed layer depth constant at 100m. The 'stratification' scenario changes only the turbulent dissipation rate.
  - Added paragraph: 'It is important to mention that, in the context of this project, in the seasonally stratified scenario only the turbulent dissipation rates varies with time, while the mixed layer depth remains constant at 100m throughout the annual cycle in all simulations.' (line 145 of revision)

- Added new paragraph in the discussion about the seasonal variability of the mixed layer depth (line 404-411 of revision)
- o Added clarification in the caption of figure 3
- **Line 296–297:** Misspelling: "remineneralization" should be corrected to "remineralization".

**Fixed**

- **Figure 9:** This is a very well-presented summary figure that nicely communicates the manuscript's story and conclusions. Therefore, it can be used as a guide while reversing manuscript`s method and result section. It also provides a helpful framework for the community. It shows the relative importance of remineralization, aggregation, fragmentation, and sinking during the three phases of carbon export. However, I was a bit confused about the arrows and Ccirt. Figure would benefit to revising of them.
  - Modified arrows. We keep the same length and the width indicates the relative importance of each process and stating that the arrows point towards the direction where mass is forced to move under the effect of each process
  - Improved captions
- Line 421: This is a good aim/paragraph and could be expanded further. The first sentence of the paragraph, in particular, would benefit from being supported by a reference for example, Henson et al. (2019), which is already included in the reference list. There is relevant literature on this topic, and incorporating additional references would help strengthen the authors' argument. In my opinion, this part of the discussion would benefit from integrating more references.

We added the proposed reference. We would like not to expand on this point though to keep the discussion concise and use all this information for the next manuscript where we couple SISSOMA to NUM plankton trait-based model.

**Reference**

Visser, A. W., Almgren, A. V., and Kandylas, A.: SISSOMA (v1): modelling marine aggregate dynamics from production to export. h

---

## Author Response (AR2)

**Comment respone**

Line 249-250: What is the relationship between machine learning techniques and the generalization of SISSOMA's architecture, and how does this enable its application in global biogeochemical models?

We added a more detailed description of how to use machine learning techniques to generalize SISSOMA which can lead to a computationally-light model able to be integrated in more sophisticated biogeochemical models.

Writing:
Line 93: Double parentheses. fixed
Lines 327 and 356: References should not be in parentheses. fixed

---

## Author Response (AR3)

**Comments about final version**

- Fixed citations
- Fixed quoting in Latex in the text
- Tidy code and update Zenodo
- Updated Zenodo link in text
- Updated acknowledgements
- Updated author contribution